# GENERALIZABLE AUTOREGRESSIVE MODELING OF TIME SERIES THROUGH FUNCTIONAL NARRATIVES

## ABSTRACT

Time series data are inherently functions of time, yet current transformers often learn time series by modeling them as mere concatenations of time periods, overlooking their functional properties. In this work, we propose a novel objective for transformers that learn time series by re-interpreting them as temporal functions. We build an alternative sequence of time series by constructing degradation operators of different intensity in the functional space, creating augmented variants of the original sample that are abstracted or simplified to different degrees. Based on the new set of generated sequence, we train an autoregressive transformer that progressively recovers the original sample from the most simplified variant. Analogous to the next word prediction task in languages that learns narratives by connecting different words, our autoregressive transformer aims to learn the Narratives of Time Series (NoTS) by connecting different functions in time. Theoretically, we justify the construction of the alternative sequence through its advantages in approximating functions. When learning time series data with transformers, constructing sequences of temporal functions allows for a broader class of approximable functions (e.g., differentiation) compared to sequences of time periods, leading to a 26% performance improvement in synthetic feature regression experiments. Experimentally, we validate NoTS in 3 different tasks across 22 real-world datasets, where we show that NoTS significantly outperforms other pre-training methods by up to 6%. Additionally, combining NoTS on top of existing transformer architectures can consistently boost the performance. Our preliminary experimental results demonstrate the potential of NoTS as a viable, theoretically justified alternative for building foundation models for time series.

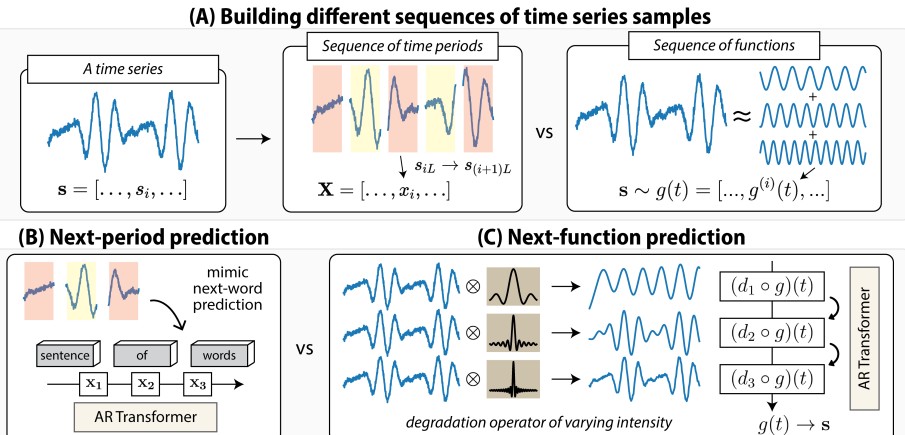

Figure 1: *Overview.* (A) Given a sample of time series, one can build different sequences from the original sample by treating it as either concatenation of time periods, or composition of temporal functions. (B) In the former case, it is common to emulate the next word prediction task in language to predict the next time period with an autoregressive (AR) transformer. (C) Alternatively, by applying degradation operators of varying intensity, we can craft augmented variants of samples that are progressively simplified, allowing a next-function prediction task. The AR transformer is trained on the alternative sequence to learn the relationship across the sequence of functions to gradually recover the variance within original samples.

# 1 INTRODUCTION

Recent advances in large language models (LLMs) demonstrate the advantage of large-scale pre-training, providing a generalizable way for modeling complex systems (Radford et al., 2019). At the core of state-of-the-art LLMs is the next-token prediction task (Achiam et al., 2023), where each data sample (sentence) is segmented into tokens (words), and the next word is predicted based on the previous words using the transformer architecture (Vaswani, 2017). By completing samples of sentences based on partial information in an autoregressive (AR) way, LLMs build generalizable data representations that can be rapidly adapted to new datasets and tasks (Brown, 2020).

Many transformer-based modeling approaches in time series analysis mimic the approaches in language by building sequences from samples through segmenting time series into periods of time points (Figure 1(A) (Nie et al., 2022; Liu et al., 2022; Zhang & Yan, 2023). An AR transformer on top of it would predict the next time period based on the existing ones (Figure 1(B) (Garza & Mergenthaler-Canseco, 2023). However, this approach has two issues: (1) slicing time series into periods **breaks nonlocal functional properties** like trend or periodicity, and often requires special remedies to compensate for the issues (Zhou et al., 2022); (2) the predicted time periods **lack generalizability**, as the prediction is sensitive to the length of chunks, the position of where the slicing happens, and the characteristics of datasets. To compensate for the issues of patching and build generalizability into transformer, recent works rely on the usage of operators like Fourier neural operators or Koopman operators, but they either require specially engineered coding blocks (Liu et al., 2023a), or a specific set of predetermined bases that may vary across datasets (Liu et al., 2024b).

Inspired by Tian et al. (2024) that replaces next-patch prediction with a next-resolution prediction task in computer vision, in this work, we re-think alternative approaches to build a coarse-to-fine sequence of time series by considering them as functions of time. Instead of slicing time series into periods, we consider time series samples $\mathbf{S}$ as a sampled version of an underlying function $g(t)$ that can be structurally simplified in its functional form (Figure 1(A)). Instead of mapping the sample onto fixed sets of basis like Taylor or Fourier series, we isolate functional components in a data-dependent way by building degradation operators $d_k(\cdot)$ of different intensity levels $k$ and progressively applying them on the signals. By doing so, we generate an alternative sequence of samples consisting of augmented variants of the signal with increasing amount of information, offering an interconnected yet simplified representation of the original signal. We train an autoregressive transformer to learn the connection of the different set of functionals, building a knowledge map of different functional components [1]. Analogous to the next word prediction task that learns narrative in languages by completing sentences, we denote our method as the Narratives of Time Series (NoTS) because it learns the functional narrative of temporal signals (Figure 1(C)).

We first justify the construction of the alternative sequence using an intuitive function approximation analysis in Section 4. When learning time series with transformers, under the universal approximation framework (Yun et al., 2019), we show that learning time series as sequences of periods of time points would cause approximation issues, as performing sampling operation on commonly encountered time series signal processing operators (e.g., differentiation) creates discontinuous sequence-to-sequence functions. Instead, such limitations can be bypassed by forming and learning from the alternative function sequences, as long as either (1) the constructed sequence is expressive, or (2) an expressive tokenizer (Ismailov, 2023) is used before learning with transformers. The analytical result is validated experimentally through a feature regression task on synthetic datasets. We show that NoTS significantly outperforms other pre-training methods when approximating features with real-world characteristics, showing its superior expressiveness both theoretically and experimentally.

We further validate NoTS in real-world time series datasets in a multitask setting, where we consider performance across 22 real-world datasets consisting of classification, imputation, and anomaly detection tasks. Across the board, NoTS improves the average performance of other pre-training methods by up to $6\%$, significantly outperforming the state-of-the-arts including next-period prediction (Garza & Mergenthaler-Canseco, 2023), masked autoencoder (MAE) (Dong et al., 2024), and MAE with Fourier neural operators (Liu et al., 2023a). Moreover, we show that NoTS can improve the performance of existing transformer architectures (Nie et al., 2022; Liu et al., 2023b), giving a consistent performance boost when performing dataset-specific pre-training. Interestingly, we present

---

[1]For example, in brain decoding tasks, signals often contain cross-frequency coupling where low-frequency components drive the high-frequency components (Klimesch, 2018; Donoghue et al., 2020).

a synthetically pre-trained lightweight model NoTS-lw, which can be efficiently adapted to real-world tasks and achieve $82\%$ average performance with only $<1\%$ parameters trained, showing the potential of NoTS on learning dynamics that can be transferred across datasets and tasks.

The main contributions of this paper are summarized as follows:

- An alternative approach to form sequences from time series data by considering them as functions of time and isolating functional components with constructed degradation operators.
- Analytical results studying transformers under the universal approximation framework, showing that learning time series from the functional perspective allows the approximation of a broader class of functions when compared to learning across periods of time.
- A novel transformer-based pre-training framework NoTS that progressively reconstruct time series from their degraded variants, and thus learn the interrelationships across functions.
- Experimental results on 2 synthetic and 22 real-world datasets, including 4 different classes of tasks, showing that NoTS significantly outperforms other pre-training methods from next-period predictors to Fourier-informed masked autoencoders, giving a stable performance boost on top of existing architectures. Preliminarily demonstrating NoTS as a viable pre-training alternative.
- A synthetically pre-trained lightweight model NoTS-lw that can be efficiently adapted on new datasets and tasks with $<1\%$ parameters trained while maintaining $82\%$ average performance.

## 2 PRELIMINARIES AND RELATED WORKS

### 2.1 PRELIMINARIES

**Autoregressive (AR) transformers** AR transformers have revolutionized natural language processing by building next-token prediction-based language models (Ray, 2023). The transformer architectures learn the interactions across different elements (tokens) in a sequence $\mathbf{X} = [\mathbf{x}_1, \mathbf{x}_2, ..., \mathbf{x}_N]$, and the AR objective is defined as follows: The probability of obtaining the next token $\mathbf{x}_i$ can be deduced from the observed subsequence $[\mathbf{x}_1, \mathbf{x}_2, ..., \mathbf{x}_{i-1}]$. Thus, the probability of obtaining the whole sample is the product of a sequence of unidirectional conditional probabilities:

$$p\left(\mathbf{x}_1, \mathbf{x}_2, \ldots, \mathbf{x}_N\right) = \prod_{i=1}^{N} p\left(\mathbf{x}_i \mid \mathbf{x}_1, \mathbf{x}_2, \ldots, \mathbf{x}_{i-1}\right) \tag{1}$$

where the AR relationship is learned by a transformer model $p_\theta$ parameterized by $\theta$. In the language domain, $\mathbf{x}_i \in \mathcal{V}$ is typically a discrete token of a word from a given vocabulary, which forms next-word predictors with impressive in-context generalization capabilities (Brown, 2020).

**Notations for time series** Time series samples are sequences of data points from multiple channels. A multivariate time series sample of $C$ channels and a length of $T$ is represented as $\mathbf{S} = [\mathbf{v}_1, \mathbf{v}_2, ..., \mathbf{v}_T] \in \mathbb{R}^{C \times T}$. Typically, each dataset has its unique channel-wise relationships.

To apply transformers on time series, one needs to form sequences from the given signal $\mathbf{S}$. A naive approach is to directly treat $\mathbf{v}_i$ as tokens and then apply transformers (Zerveas et al., 2021). The drawback is that the token representation space is dependent only on $\mathbf{v}_i$, which varies across datasets, making it less generalizable. Recently, many time series framework produce tokens through cutting time series into different periods of time with a length of $L$, which creates tokens $\mathbf{x}_i = \text{Tokenizer}([\mathbf{v}_{iL}, ..., \mathbf{v}_{(i+1)L}])$ that contain more dynamics (Ren et al., 2022). To further eliminate the negative impact of channel-wise relationships on generalizability, Nie et al. (2022); Liu et al. (2022) considers the channel-independent design, which processes each channel (row) of $\mathbf{S}$ independently, producing tokens based on individual channels for transformers. While the approach demonstrated more generalizability, it is computationally expensive in high-density settings, which was later discussed by other works (Zhang & Yan, 2023).

### 2.2 PRETRAINING METHODS FOR TIME SERIES

Pre-training on large-scale datasets has proven effective in helping models learn generalizable patterns, which is particularly advantageous in the time series domain where downstream datasets are often small-scale (Liu et al., 2021; Zhang et al., 2022; Woo et al., 2022). There are two prominent approaches for reconstruction-based pre-training in transformers: masked modeling and next-period prediction. Masked modeling trains transformers by randomly masking elements, and predicting the masked values with the remaining sequence. Representative works include SimMTM (Dong et al., 2024), which implements masked modeling through aggregating neighboring points, and bioFAME

(Liu et al., 2023a), which employs Fourier-based kernels to achieve the same objective. However, these methods suffer from the loss of nonlocal information. Several approaches are proposed to mitigate this issue, such as leveraging multi-resolution patches (Das et al., 2023; Woo et al., 2024). More recently, the next-period prediction approach has gained attention for pre-training, particularly in the development of foundation models. For instance, Time-GPT1 (Garza & Mergenthaler-Canseco, 2023) implements next-period prediction in a straightforward manner, while Chronos (Ansari et al., 2024) applies scaling and quantization techniques to tokenize time series data and model the categorical distributions. Despite their success, these models also suffer from the challenge of losing nonlocal information, which is partially addressed through the use of lagged features and temporal covariates in Lag-Llama (Rasul et al., 2023). Based upon previous works, our work aims to fundamentally address the issue through building and learning sequences from the function perspective.

An alternative line of research seeks to adapt language models directly for time series applications. Some approaches transfer pre-trained weights from LLMs and retrain the tokenization layers to handle time-series-specific tasks (Zhou et al., 2023; Cao et al., 2023; Liu et al., 2024c). Another line of work focuses on reprogramming time series data into text, and applying LLMs to process the textual inputs with time series prompts (Jin et al., 2023; Xue & Salim, 2023). While these works primarily aim to bridge the modality gap between LLMs and time series applications, NoTS is specifically designed for time series to capture the subtle variations and dynamics inherent in temporal signals.

### 2.3 LEARNING TIME SERIES FROM THE FUNCTIONAL PERSPECTIVE

A line of traditional time series modeling methods focus on learning samples from the functional perspective (Chapados & Bengio, 2007), using statistical approaches (Holt, 2004), or their advanced variations like the Theta method (Assimakopoulos & Nikolopoulos, 2000) or ARIMA models (Hyndman & Khandakar, 2008). These methods have been extended to deep learning through basis expansion approaches, with N-BEATS (Oreshkin et al., 2019) being a prominent example. N-BEATS uses fully connected layers to perform hierarchical time series decomposition by generating coefficients for predefined or learnable neural bases. N-HiTS (Challu et al., 2023) builds on N-BEATS by incorporating subsampling layers, enabling multifrequency data sampling and multi-scale interpolation for improved predictions. Our work differs from prior approaches by focusing on the advantages of building sequences with functional awareness through the transformer architecture.

## 3 METHODS

Alternative to modeling time series as sequences of fragmented time periods, our framework is built on the idea to model time series as sequences of constructed temporal functions with transformers. We begin by introducing the high-level objective in Section 3.1, and then introduce the pre-training method NoTS in Section 3.2 as well as how to adapt it in real-world tasks in Section 3.2.

### 3.1 THE NEXT-FUNCTION PREDICTION TASK

We assume that each signal $\mathbf{S} = [\mathbf{v}_1, \mathbf{v}_2, ..., \mathbf{v}_T] \in \mathbb{R}^{C \times T}$ is intrinsically controlled by a temporal function $g(t) : \mathbb{R} \to \mathbb{R}^C$, where $\mathbf{v}_i = g(i)$ is the product of a sampling process in time. To train a transformer with awareness of the functional perspective, we build sequences of functions, where each element is a simplified version of the original sample. Practically, we ask two questions:

- How to build meaningful functional elements, as they tend to change across different datasets?
- How to form a meaningful sequence for the transformer, so that we can enforce the transformer to learn generalizable representation from the constructed sequences of functional components?

In this work, we propose to construct *degradation functions* $d_k(\cdot)$ of intensity $k$, that generate augmentations of the original function $g(t)$ with varying levels of partial information. Applying degradation functions $d_k(\cdot)$ on signals generates data-dependent functions as tokens for transformer, removing the need for a fixed set of bases. By controlling the intensity of degradation, we create $g_k(t) = (d_k \circ g)(t)$, where $g_{k+1}(t)$ contains strictly more or an equal amount of information than $g_k(t)$ about the original sample $g(t)$, establishing a progressive relationship for the transformer to learn. Based on the constructed sequence, the new modeling approach becomes the following:

$$p\left(g_1(t), g_2(t), \ldots, g_K(t)\right) = \prod_{k=1}^{K} p\left(g_k(t) \mid g_1(t), g_2(t), \ldots, g_{k-1}(t)\right), \qquad (2)$$

where $K \to \infty$, $\mathbf{v}_i = g_\infty(i)$ forms the actual time series $\mathbf{S}$ under the sampling operation.

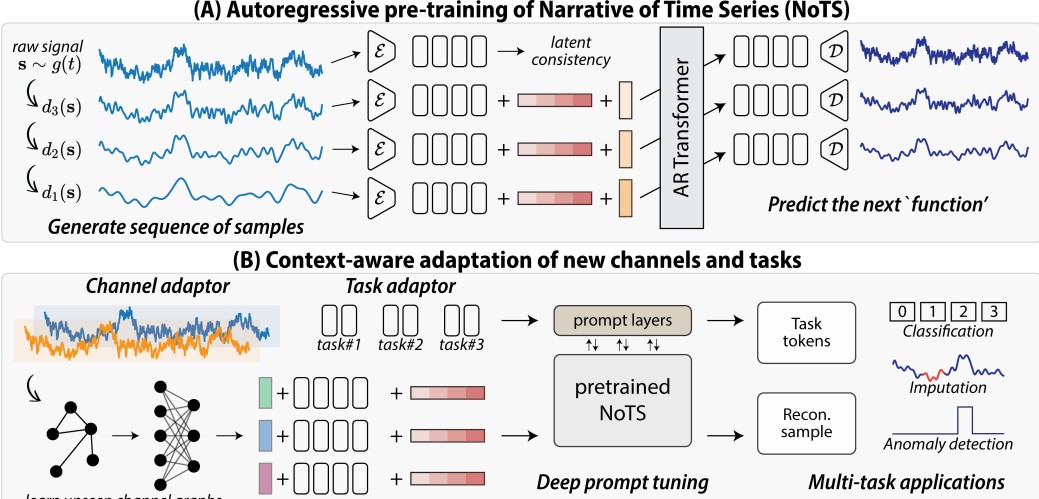

Figure 2: *Narrative of Time Series (NoTS)*. (A) To perform autoregressive pre-training of NoTS, we first generate a sequence of time series from the raw signal that progressively simplifies the sample. The generated signals are passed into an encoder, added with position and resolution embeddings before fed into the AR transformer, which is trained with a decoder to reconstruct the signal of the next resolution. The raw signal was passed into a latent consistency loss directly. (B) To apply a pre-trained model on real-world dataset, we construct channel adaptor and task adaptor that handles unseen channel graphs and new tasks, respectively. The channel adaptor consists of a multi layer perceptron that pre-process channel maps and new additive channel embeddings. The task adaptor is newly initialized tokens that are prompted into the transformer following Jia et al. (2022). The produced task tokens and reconstructed samples are later used in multitask applications through this context-aware adaptation pipeline.

### 3.2 NoTS: A NOVEL PRE-TRAINING OBJECTIVE FOR TRANSFORMERS

Based on the framework, the pre-training objective NoTS consists of the following components.

**Local and global degradation functions**    Practically, we construct degraded signals with convolution operators: $\mathbf{S}_k = d_k(\mathbf{S}) = (\mathbf{S} * w_k)[n]$, where $*$ represents discrete convolution between rows of signal $\mathbf{S}$ and a kernel $w_k$. We use two different kernels as defined below:

- *Local smoothing:* A simple averaging kernel of different lengths $p_k$ is used for local smoothing. Specifically, $w_k[n] = 1/p_k$ for $-0.5p_k \leq n \leq 0.5p_k$ and $w_k[n] = 0$ elsewhere. The set of even numbers $\{p_k\}$ is selected as hyperparameters with descending order as $k$ increases.
- *Global smoothing:* A low-pass filter with different frequency cutoff values is used for global smoothing. Specifically, we build $w_k[n] = \text{sinc}(p_k n)$, where $\{p_k\}$ is a set of values that control the frequency cutoff of $0.5p_k$ as hyperparameters with descending order as $k$ increases.

By constructing both local and global smoothing degradation functions, the proposed method can simultaneously model autoregressive relationships across different smoothness and frequencies, covering a prevalent range of tasks in both time and frequency.

**Autoregressive modeling of groups of tokens in the latent space**    We build tokenizers to convert the constructed signals $\mathbf{S}_k$ into recognizable embeddings for transformers. Ideally, we want to use one token for each function, yet this is computationally infeasible as the length of signals increases. Instead, we rely on existing encoder/decoder architectures as tokenizers to convert signals into group of tokens in the latent space to perform AR modeling. Specifically, the encoder produces groups of tokens from each signal $\mathbf{R}_k = \mathcal{E}(\mathbf{S}_k)$ and the decoder reconstructs signals from groups of tokens $\mathbf{S}'_k = \mathcal{D}(\mathbf{R}'_k)$, where each $\mathbf{R}_k$ and $\mathbf{R}'_k$ consists of multiple tokens for the transformer. The AR modeling is enforced by applying group-wise masking to the transformer attention map, where:

$$[\mathbf{R}'_2, ..., \mathbf{R}'_K] = \text{Transformer}([\mathbf{R}_1, ..., \mathbf{R}_{K-1}]), \text{ with } \text{mask}[\Omega_k] = \begin{cases} 0, & \bigcup_{m=1}^k \Omega_m \\ -\infty, & \text{elsewhere} \end{cases}$$

and $\Omega_k$ represents the set of sequence positions corresponding to $\mathbf{R}_k$. While $\mathcal{E}$ and $\mathcal{D}$ can be any encoder/decoder architectures, we implement a lightweight model NoTS-lw with a simple channel-independent 1D-ResNet encoder/decoder block to maintain fidelity in the token space. We also report results with different encoder/decoder architectures in Table 2.

**Positional embeddings**  Since transformers are inherently invariant to the order of sequences, it is critical to embed sufficient information about the raw position into the transformer to ensure $\mathbf{R}_k$ includes sufficient information about samples. Thus, we add the following embeddings:

- *Group embeddings*. To help transformers learn sufficient information about each function embedded by $\mathbf{R}_k$, we apply rotary positional embedding (Su et al., 2024) on each group of tokens to encode the relative sequence position within each group or cohort of tokens.
- *Degradation embeddings*. We add learnable absolute positional embeddings that encode the relative degree of degradation of each augmented variant of signals. In other words, the degradation embeddings encode information $k$ of the degradation function $d_k(\cdot)$.
- *Channel embeddings (optional)*. When applying channel-independent architectures as the encoder, we add an additional set of learnable absolute positional embeddings along with the group embeddings to help encode channel-wise relationship within each group of tokens.

**Training objective**  We perform a self-supervised autoregressive reconstruction task to learn meaningful representations of time series using the proposed framework. To achieve this, we minimize the differences between $\mathbf{S}_k$ and the reconstructed $\mathbf{S}'_{k+1}$ for every $k < K$. If only the AR loss is considered, the latent of the raw data $\mathbf{R}_K$ would remain unused by encoder/decoder throughout the training process. To avoid the resulting distributional shift, we add a *latent consistency* term, that regularizes the consistency between the latent of the raw data to be able to be reconstructed back. Thus, for each input signal matrix $\mathbf{S} \in \mathbb{R}^{C \times T}$, the training optimizes the following loss:

$$\mathcal{L} = \sum_{k=1}^{K-1} \underbrace{\mathcal{L}_{\text{recon}}(\mathbf{S}'_{k+1}, \mathbf{S}_k)}_{\text{AR reconstruction}} + \underbrace{\mathcal{L}_{\text{recon}}(\mathcal{D}(\mathcal{E}(\mathbf{S}_K)), \mathbf{S}_K)}_{\text{latent consistency term}} \tag{3}$$

where $\mathcal{L}_{\text{recon}}$ is the reconstruction loss (mean absolute error is used throughout the paper).

## 3.3 MODEL DEPLOYMENT PIPELINE AND CONTEXT-AWARE ADAPTATION

As a pre-training strategy, NoTS considers a pre-training dataset $D_{\text{PT}}$ and a downstream dataset $D_{\text{FT}}$ with sample dimensions $C_{\text{PT}} \times T_{\text{PT}}$ and $C_{\text{FT}} \times T_{\text{FT}}$, respectively, allowing for differing channel and temporal dimensions between these two phases (Liu et al., 2023a; Dong et al., 2024). To deploy the model, we first pre-train it by performing the autoregressive reconstruction task, guided by the training objective as detailed in Equation 3. The pre-trained model is later fine-tuned on the training split of the downstream dataset $D_{\text{FT}}$, and is finally evaluated on the testing split of $D_{\text{FT}}$. In this work, we are primarily interested in adapting a pre-trained NoTS model in a context-aware way with the help of the following two adaptors on new channel maps and tasks:

**Channel adaptors**  To learn new channel-wise relationship at test time, we build channel adaptors as follows: (1) To encourage information exchange across channels, we add a data embedding layer before applying the encoder $\mathcal{E}$. The data embedding layer is a simple linear layer that encodes on the channel dimension $\mathbb{R}^C \to \mathbb{R}^{C'}$ to mix channel-wise information at an early stage. (2) We also re-initialize and re-train additive channel tokens for each dataset when applicable.

**Task adaptors**  To apply the pre-trained models on a diverse set of tasks, we build task adaptors as follows: (1) We initialize and append prompt tokens to the transformer architecture along with data tokens following the deep visual prompt tuning plan as detailed in Jia et al. (2022); (2) We also add task-specific linear layers at the end of the transformer for inference purpose.

Given the two adaptors, we can perform context-aware adaptation of NoTS on new channel maps and tasks, allowing the transfer of general-purpose dynamics that are learned at the pre-training stage. Interestingly, the adaptation pipeline is also parameter-efficient: The adaptors add $<1\%$ new parameters in comparison to the original model consists of encoder, transformer, and decoder.

Beyond context-aware adaptation, in this work, we comprehensively evaluate NoTS in both cross-domain and within-domain settings (Appendix C.1.2), compare the models' performance using full-scale model fine-tuning and prompt tuning schemes (Appendix C.1.3), and apply NoTS on various downstream tasks (Appendix C.3). We refer the readers to Appendix for technical details.

# 4    AN INTUITIVE EXAMPLE: APPROXIMATING FUNCTIONS

To justify the construction of functional sequences, we provide an intuitive example to investigate the expressive power of transformers under the context of time series domain. Following Yun et al. (2019); Luo et al. (2022), we consider the standard transformer architectures:

$$\mathcal{T}^{h,m,r} := \left\{ f : \mathbb{R}^{d \times n} \to \mathbb{R}^{d \times n} \mid f \text{ consists of Transformer blocks } t^{h,m,r} \right\}.$$

where $t^{h,m,r}$ consists of one self-attention layer of $h$ heads of size $m$ and one feed-forward layer with $r$ hidden dimensions (see definitions in Appendix Eq. 6). To remove the restriction of permutation equivariance, we consider adding absolute positional embedding[2] to the transformer that creates $\mathcal{T}_P^{h,m,r} := \{ f_P(\boldsymbol{X}) = f(\boldsymbol{X} + \boldsymbol{E}) \}$ where $f \in \mathcal{T}^{h,m,r}$ and $\boldsymbol{X}, \boldsymbol{E} \in \mathbb{R}^{d \times n}$. Detailed in Appendix A, our analysis is an extension of previous results in Yun et al. (2019); Ismailov (2023).

## 4.1    TIME SERIES IN THE FUNCTION SPACE

This work assumes time series data as functions in time $g(t)$, which forms function space $\mathcal{F}_g(\mathbb{R})$. An operator on the function space maps the original function $g(t)$ to a target function $h(t)$, creating a mapping across function spaces $A : \mathcal{F}_g(\mathbb{R}) \to \mathcal{F}_h(\mathbb{R})$. Assume that the signal is produced with a sampling plan $\{t_i\}_{i=1}^T$, the sampling operation on top of the functions discretizes the mapping into a set of output $\{A[g(t_i)]\}_{i=1}^T$, which forms a sequence-to-sequence function $f_{(A)} : \mathbb{R}^{d \times T} \to \mathbb{R}^{d \times T}$. When breaking time series into concatenations of time periods, $\mathbf{S}$ is directly treated as inputs to the transformer $\mathbf{X}$, where one aims to find a transformer network $f_P \in \mathcal{T}_P^{h,m,r}$ to approximate $f_{(A)}$.

**Example: The differential operator**    It is intuitive that one can easily construct a linear but discontinuous mapping $A$, which is not necessarily approximable by transformers. See below:

**Theorem 1.** Given $T > 2$, and $\mathcal{D} \subseteq \mathbb{R}^{d \times T}$. Consider the differential operator $A$ that forms a sequence-to-sequence function $f_{(A)}$ under sampling plans $\{t_i\}_{i=1}^T$ with its initial starting point $t_1 \in \mathbb{R}$ and fixed intervals. There exists a $\mathbf{X} \in \mathbb{R}^{d \times T}$, such that:

$$\sup_{\boldsymbol{X} \in \mathcal{D}} \left\| f_P(\boldsymbol{X}) - f_{(A)}(\boldsymbol{X}) \right\|_2^2 \geq T \tag{4}$$

holds for any transformer network $f_P \in \mathcal{T}_P^{h,m,r}$.

**Proof.** We construct a negative example with $d = 1$. Consider a set of input functions $g_M(t) = \sin(Mt)/M$, the target functions under the differential operator are $h_M(t) = \cos(Mt)$. As $M$ increases, the input function converges uniformly to a constant zero function, which gives a sampled input matrix $\boldsymbol{X} \to \boldsymbol{0} \in \mathcal{D}$. At limit, the studied transformer network $f_P(\boldsymbol{X})$ converges to a fixed matrix $\mathbf{B}$ (see Appendix A.2). Thus, given a sampling plan of interval $t_{i+1} - t_i = \pi/M$ and two initial starting points $t_1^{(1)} = 0$ and $t_1^{(2)} = \pi$, we form $\boldsymbol{X}_1$ and $\boldsymbol{X}_2$ that give:

$$\lim_{M \to \infty} \left\| f_P(\boldsymbol{X}_1) - f_{(A,M)}(\boldsymbol{X}_1) \right\| + \left\| f_P(\boldsymbol{X}_2) - f_{(A,M)}(\boldsymbol{X}_2) \right\| \geq \sum_{i=1}^T 2 = 2T \tag{5}$$

where $f_{(A,M)}$ denotes the function formed from $\{A[g_M(t_i)]\}_{i=1}^T$, which leads to Eq. 4.

## 4.2    TWO SUFFICIENT CONDITIONS TO APPROXIMATE THE DIFFERENTIAL OPERATOR

When considering signals as functions in time, sampling from simple signal processing operators may create discontinuous sequence-to-sequence functions, causing approximation issues of transformer if one directly considers $\mathbf{S}$ as inputs $\mathbf{X}$ to the transformer. Instead, by constructing signals sequences of length $T$ using $\mathbf{S}_k = d_k(\mathbf{S})$, and performing dimensionality reduction with an encoder $\mathcal{E}$, we create two sufficient conditions to address the approximation issue as follows:

**Proposition 1.** Given a signal $\mathbf{S} \in \mathbb{R}^{d \times T}$ and an encoder $\mathcal{E} : \mathbb{R}^{d \times T} \to \mathbb{R}^d$, there exists two sufficient conditions to approximate $\{A[g(t_i)]\}_{i=1}^T$ with the construction of $\mathbf{X} = [\mathcal{E}(\mathbf{S}_1), \mathcal{E}(\mathbf{S}_2), ..., \mathcal{E}(\mathbf{S}_T)]$:

- The constructed $\mathbf{S}_i$ is expressive such that there exists a continuous mapping between a fixed element of $\mathbf{S}_i$ and the $i$-th element of the target output $A[g(t_i)]$;
- Given any distinguishable $\mathbf{S}_i$, there exists an expressive tokenizer $\mathcal{E}$ that preprocess $\mathbf{S}_i$ to create a continuous mapping between $\mathcal{E}(\mathbf{S}_i)$ to the target.

**Proof and examples.** See Appendix A.3 for proof and an example solution for differential operator.

---

[2]Refer to Luo et al. (2022) for a case study of relative positional embedding under the UA framework.

Table 1: Feature approximation results on synthetic datasets. We compare the function approximation ability of different pre-training methods given the same architecture and pre-training pipeline. All presented numbers are averaged across three runs and scaled by 100 for better readability. Lower numbers are better.

| | Fractional Brownian motion (fBm) | | | Autocorrelated sinusoids | | |
|---|---|---|---|---|---|---|
| Regression ($\downarrow$) | $\mathcal{H}$-index (1D) | SSC (32D) | WAMP (32D) | SSC (32D) | WAMP (32D) | b. power (96D) |
| VQVAE | $3.78 \pm 0.45$ | $38.93 \pm 0.70$ | $65.77 \pm 3.72$ | $26.24 \pm 0.61$ | $29.13 \pm 0.90$ | $14.37 \pm 0.03$ |
| MAE | $2.01 \pm 0.61$ | $25.78 \pm 0.11$ | $26.34 \pm 0.03$ | $25.29 \pm 0.31$ | $28.81 \pm 2.86$ | $14.90 \pm 0.02$ |
| FAMAE | $1.99 \pm 0.24$ | $33.85 \pm 0.53$ | $45.76 \pm 0.24$ | $28.26 \pm 0.57$ | $24.82 \pm 0.84$ | $13.92 \pm 0.02$ |
| Next-period pred. | $1.75 \pm 0.11$ | $27.38 \pm 0.12$ | $26.66 \pm 0.19$ | $24.44 \pm 0.11$ | $28.97 \pm 1.37$ | $13.96 \pm 0.04$ |
| **NoTS (Ours)** | $1.27 \pm 0.16$ | $23.78 \pm 0.34$ | $20.04 \pm 0.12$ | $23.13 \pm 0.19$ | $24.58 \pm 0.48$ | $13.62 \pm 0.05$ |
| Improvement | $\uparrow \mathbf{37.80\%}$ | $\uparrow \mathbf{8.41\%}$ | $\uparrow \mathbf{31.44\%}$ | $\uparrow \mathbf{5.66\%}$ | $\uparrow \mathbf{0.98\%}$ | $\uparrow \mathbf{2.20\%}$ |

## 5 EXPERIMENTAL RESULTS

While the approximation analysis posts strong assumptions on the solution including the minimal length $T$ of the constructed sequence and the use of specific encoder $\mathcal{E}$. In this section, we show that, experimentally, NoTS works in both synthetic and real-world applications with relaxed assumptions.

### 5.1 SYNTHETIC EXPERIMENTS: A FEATURE REGRESSION TASK

**Datasets** We build two synthetic datasets with AR components in time and frequency spaces.

(1) Fractional Brownian motion (fBm). fBm is a generalized Gaussian process with special covariance structure that was found to be similar to many types of time series datasets such as traffic, stock prices, and biosignals (Rivero et al., 2016). Unlike the classic Brownian motion, fBm has interdependent increments across time that are controlled by the Hurst index $\mathcal{H} \in (0, 1)$, which creates autoregressive components in time that exhibit long-range ($\mathcal{H} > 0.5$) or short-range ($\mathcal{H} < 0.5$) dependencies. We simulate the fBm process 20,000 times to create signals of length 1024 using the Cholesky's Method (Dieker & Mandjes, 2003) with $O(l^3)$ complexity, and remove the generated signals with abnormal values due to simulation instability (around 0.5% of all data).

(2) Superposition of autocorrelated sinusoids. We extend previous synthetic datasets based on sinusoids (Yoon et al., 2019; Das et al., 2023), and build a new synthetic dataset of sinusoids with AR components in the frequency space. Specifically, we sample the set of $\{f_i\}_{i=1}^B$ based on five random AR(B/2) processes, where we set $B = \{20, 16, 10, 8, 4\}$. We generate amplitude following the $a_i = 1/f_i$ frequency distribution, uniformly sample the phase $p_i \in (0, 2\pi]$, and add Gaussian noise $0.05 * \mathcal{N}(0, 1)$ to the signal. We randomly initialize each process 10,000 times, creating a dataset of 50,000 samples where each sample is of length 1024.

**Results on feature regression** We estimate different pre-training methods' capability of approximating functions with the feature regression task. The ground truth of features are built based on common signal processing analysis methods Slope Sign Change (SSC, 32D) and Willison Amplitude (WAMP, 32D), and we also include the Hurst index ($\mathcal{H}$-index, 1D) for the fBM dataset and the band power (b. power, 96D) for the sinusoids (Appendix B.1). Note that SSC and WAMP are both implemented with global thresholding, making them discontinuous sequence-to-sequence functions. Following Section 3.2, we train a VQVAE (Van Den Oord et al., 2017), masked autoencoder (MAE) (Dong et al., 2024), frequency-aware MAE (FAMAE) (Liu et al., 2023a), next-period prediction transformer (Garza & Mergenthaler-Canseco, 2023), and **NoTS-lw** on the synthetic datasets by appending them with a regression task adaptor to validate the performance of our proposed method.

As shown in Table 1, across the board, **NoTS-lw** significantly outperforms all other pre-training methods given the same architecture and training pipeline. The relative improvement is especially pronounced in the fBm dataset, where data has complicated covariance architecture that was found relevant in many real-world applications, where we have 26% improvements across the features.

**Visualizing the next-function prediction process** We present the data and latent visualizations in Figure 3. In Figure 3(A), we show the original data sequence $\{\mathbf{S}_i\}_{i=1}^K$ and the reconstructed data sequence $\{\mathbf{S}'_i\}_{i=2}^K$. Note that the original signal $\mathbf{S}_K$ was not passed into the transformer. We can see that the predicted sequence has information that is not presented in previous signals, showing the function prediction capacity of the transformer. In Figure 3(B), we plot the token space before or after the AR transformer using the PCA reduction on $\{\mathbf{R}_i\}_{i=1}^K$ and $\{\mathbf{R}'_i\}_{i=2}^K$, respectively. When coloring the tokens differently based on their degradation parameter $i$, we observe that: (1) In origi-

Table 2: Comparisons between NoTS and other pre-training methods on real-world datasets. We categorize the results based on (a) if adaptors are used, and (b) if the weights of the pre-trained models are frozen. We compute an average error rate ($\downarrow$) to compare the final performance of different methods in each condition.

| Methods | (a) | (b) | *Classification* ($\uparrow$) | | *Anom. Det.* ($\uparrow$) | | | | *Imputation* ($\downarrow$) | | | | **Avg.** ($\downarrow$) |
| | | | UCR-9 | UEA-5 | SMD | MSL | SWaT | PSM | ETTm1 | ETTm2 | ETTh1 | ETTh2 | **error rate** |
|---|---|---|---|---|---|---|---|---|---|---|---|---|---|
| SimMTM | ✓ | ✓ | 68.70 | 55.36 | **84.06** | 83.90 | 91.20 | 96.07 | **0.164** | 0.126 | 0.264 | 0.183 | 19.43 |
| bioFAME | ✓ | ✓ | 62.63 | 60.32 | 83.09 | **84.28** | 91.21 | 95.94 | 0.203 | **0.122** | **0.258** | **0.178** | 19.87 |
| Next-pred | ✓ | ✓ | 65.95 | 58.30 | 82.96 | 83.75 | 90.47 | **96.54** | 0.306 | 0.178 | 0.465 | 0.270 | 24.39 |
| **NoTS-lw (Ours)** | ✓ | ✓ | **71.88** | **62.78** | 83.63 | **84.28** | **93.26** | 96.27 | **0.164** | 0.126 | 0.286 | 0.196 | **18.51** |
| SimMTM | ✓ | ✗ | 81.65 | 61.23 | 83.48 | 84.11 | 91.35 | 96.36 | 0.123 | **0.107** | **0.201** | 0.166 | 16.14 |
| bioFAME | ✓ | ✗ | 81.53 | 63.57 | 83.59 | 83.98 | **91.46** | 96.88 | 0.129 | **0.107** | 0.202 | 0.178 | 16.05 |
| Next-pred | ✓ | ✗ | 80.62 | 62.76 | 83.00 | 84.09 | 91.00 | 96.87 | 0.130 | 0.119 | 0.228 | 0.188 | 16.82 |
| **NoTS-lw (Ours)** | ✓ | ✗ | **88.08** | **66.38** | 84.19 | 84.15 | 91.26 | 96.88 | **0.122** | 0.116 | 0.218 | **0.163** | **15.10** |
| PatchTST | ✗ | ✗ | 83.57 | 63.31 | 78.96 | 78.81 | 83.75 | 78.07 | 0.181 | 0.126 | 0.347 | 0.187 | 21.78 |
| **+NoTS (Ours)** | ✗ | ✗ | ↑**1.71** | ↑**1.65** | ↑**2.20** | ↑**3.96** | ↑**5.97** | ↑**11.25** | ↓**.003** | ↓**.003** | ↓**.064** | ↓**.006** | **18.33** |
| iTransformer | ✗ | ✗ | 82.67 | 67.62 | 85.18 | 83.04 | 91.88 | 97.07 | 0.162 | 0.111 | 0.240 | 0.168 | 16.07 |
| **+NoTS (Ours)** | ✗ | ✗ | ↑**1.26** | ↑**0.65** | ↑**0.17** | ↑**0.11** | ↑**0.05** | ↑**0.01** | ↑**.005** | ↓**.002** | ↓**.013** | ↓**.004** | **15.70** |

nal token space $\{\mathbf{R}_i\}_{i=1}^{K}$, severely degraded signals generates more clustered tokens, and the tokens would gradually disperse as signals become more realistic; (2) The predicted tokens $\{\mathbf{R}_i'\}_{i=2}^{K}$ would generate a token space with similar behaviour without seeing the original set of tokens $\mathbf{R}_K$. This behaviour demonstrates the autoregressive capacity of the transformer.

## 5.2 REAL-WORLD EXPERIMENTS: CONTEXT-AWARE GENERALIZATION

**Experimental setups**   To examine the performance of NoTS in real-world applications, we perform multi-task validation following the setups in Wu et al. (2022). Specifically, we perform the classification task on the UCR subset (Dau et al., 2019) and UEA subset (Bagnall et al., 2018); the imputation task on the ETDataset (Zhou et al., 2021), and the anomaly detection task on MSL (Hundman et al., 2018), PSM (Abdulaal et al., 2021), SWaT (Mathur & Tippenhauer, 2016), and SMD (Su et al., 2019) datasets. We follow Wu et al. (2022) for standard data pre-processing and task deployment pipeline, except for the imputation task where we tested a more challenging variant of channel-wise imputation (see Appendix B.2.2 and B.2.3 for details and original imputation results).

To validate that NoTS is a superior pre-training method, we perform two sets of experiments: First, we compare the performance of NoTS-lw against the next-period AR transformer, a MAE (Dong et al., 2024), and a frequency-aware MAE (Liu et al., 2023a) by pre-training them on synthetic datasets and deploying the prompt tuning pipeline for all pre-trained base models. Second, we append the pre-training pipeline NoTS on top of existing architectures PatchTST (Nie et al., 2022) and iTransformer (Liu et al., 2023b), and compute the performance benefits from adding NoTS.

**Experimental results**   As shown in Table 2, with or without parameters frozen, NoTS-lw significantly outperforms all other pre-training methods. Specifically, given the same pre-training pipeline and architecture, NoTS-lw outperforms other method across all tasks by up to 6% average. Interestingly, we note that NoTS-lw show comparable performance on imputation tasks, where MAE-like architectures are trained to perform the task. Additionally, when attaching NoTS on existing architectures PatchTST (Nie et al., 2022) and iTransformer (Liu et al., 2023b), NoTS improves their performance without specific backbone or adaptors, showing the versatility of the pre-training method.

Interestingly, we should like to emphasize on the context-aware generalization ability of NoTS. With the architecture frozen (first 4 rows of Table 2), we only train <1% of the parameters, yet it performs 82% performance, potentially demonstrating the context-aware generalization.

## 5.3 ABLATION EXPERIMENTS AND MODEL ANALYSIS

**Ablation of effective components in NoTS**   In Table 3, we perform ablations of NoTS by isolating the effective components of NoTS-lw in the feature regression task (the $\mathcal{H}$-index). Specifically, we train three variants of NoTS-lw by (1) removing the latent consistency term in training loss, (2) removing the autoregressive masking within transformer, creating a transformer that merely bridges tokens of the augmented samples, (3) removing the connections among constructed augmentations, and using degradation operator only as augmentations. As expected, removing the latent consistency term would cause distributional shift as the model never sees raw data, and would result in severe

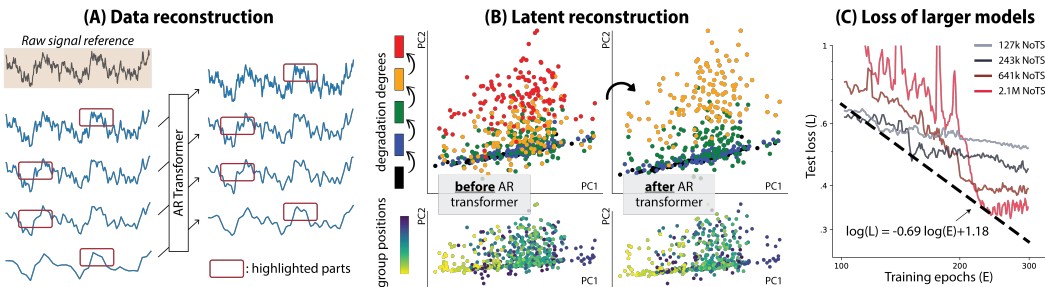

Figure 3: *Visualizations of AR performance and loss.* (A) We visualize the autoregressive inference process of NoTS on the synthetic dataset. From bottom to top, the signal variance is gradually recovered through the prediction of the AR transformer. (B) The token space is visualized through principal component analysis, where tokens of the simplified signals gradually disperse to a larger region when colored in different degradation degrees. When colored with relative group positions, the distribution does not shift as much on the direction of another principal component. (C) A pilot study shows that training larger NoTS models leads to lower reconstruction loss on the test set, potentially following the power law behaviour of AR models.

performance degradation. Interestingly, training a transformer that connects augmented samples can also provide improved performance, as observed in other works (Hou et al., 2022; Liu et al., 2024a).

**Connection to diffusion models**   One might relate our work with diffusion models (Ho et al., 2020) by using a stochastic additive Gaussian noise of varying degrees as a degradation operator. We attempted this as model variant (4) in Table 3, yet the performance is inferior to the convolution-based degradation operators. One hypothesis is that time series data is inherently noisy, and adding Gaussian noise instead of performing smoothing or filtering can be less effective, as observed in audio signals (Dieleman, 2024). Building connections between NoTS and cold diffusion models with deterministic degradation operators (Bansal et al., 2024), or more recent diffusion models (Chen et al., 2023) can be an exciting future research direction.

Table 3: Ablation experiments. Columns denote: If the model sees original signal (orig.), if the model uses autoregressive modeling (AR), if the signal variants are connected (conn.), and if a Gaussian-based degradation operator is used ($d_k^{(\mathcal{N})}$).

|      | orig. | AR | conn. | $d_k^{(\mathcal{N})}$ | error ($\downarrow$) |
|------|-------|-----|-------|-----------------------|----------------------|
| (1)  | ✗     | ✓   | ✓     | ✗                     | 1.75                 |
| (2)  | ✓     | ✗   | ✓     | ✗                     | 1.48                 |
| (3)  | ✓     | ✗   | ✗     | ✗                     | 1.82                 |
| (4)  | ✓     | ✓   | ✓     | ✓                     | 1.69                 |
| NoTS | ✓     | ✓   | ✓     | ✗                     | 1.27                 |

**Scalability analysis**   While this work aims only to provide an initial experimental exploration of the proposed pre-training methodology NoTS, we attempted a pilot study to increase the size of NoTS-lw to demonstrate its potential given more parameters. We trained four models with 127k, 243k (used in all previous experiments), 641k, 2.1M parameters to observe their performance. As shown in Figure 3(C), when fixing the amount of training data, training the models to convergence with increased parameters leads to increased performance, potentially following a power law curve of AR frameworks in language and computer vision (Kaplan et al., 2020; El-Nouby et al., 2024).

## 6    CONCLUSION

In this paper, we propose a novel autoregressive pre-training method NoTS for time series. Our work aims to provide an alternative view of time series by considering them as functions of time instead of concatenations of time periods. This novel perspective allows us to construct degradation operators, which build an alternative sequence as inputs to the transformer. The transformer is pre-trained with an autoregressive loss to encourage the learning of cross-function relationship, building a model that can recover signal variability from their simplified variants. We validated the performance of NoTS with experimental results on 2 synthetic and 22 real-world datasets, demonstrating its superiority among existing pre-training methods across multiple tasks, showing a viable alternative for developing foundation models for time series analysis in the future.

**Limitations**   Future works may extend the existing results through: (1) Expanding our initial experimental efforts to larger models, larger-scale datasets, and more challenging tasks. (2) Building in-depth theoretical connection to diffusion-based models, connecting NoTS with recent works (Dieleman, 2024) from the audio and computer vision domain. (3) Understanding how NoTS performs in stochastic events as detailed in (Kidger et al., 2020) based on rough path theory.

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
