APPENDIX

# A ANALYTICAL RESULTS

## A.1 PRELIMINARIES

**Transformers** We consider the standard transformer architecture as defined in (Luo et al., 2022). The transformer network is the stack of transformer blocks, each of them consists of a self-attention layer $\text{Attn}(\cdot)$ and a feed forward layer $\text{FF}(\cdot)$. Given an input $\boldsymbol{X} \in \mathbb{R}^{d \times T}$, they are written as:

$$\text{Attn}(\boldsymbol{X}) = \boldsymbol{X} + \sum_{i=1}^{h} \boldsymbol{W}_O^i \boldsymbol{W}_V^i \boldsymbol{X} \cdot \sigma \left[ \left( \boldsymbol{W}_K^i \boldsymbol{X} \right)^T \boldsymbol{W}_Q^i \boldsymbol{X} \right] \tag{6}$$

$$\text{FF}(\boldsymbol{X}) = \text{Attn}(\boldsymbol{X}) + \boldsymbol{W}_2 \cdot \text{ReLU} \left( \boldsymbol{W}_1 \cdot \text{Attn}(\boldsymbol{X}) \right)$$

where $\boldsymbol{W}_O^i \in \mathbb{R}^{d \times m}$, $\boldsymbol{W}_V^i, \boldsymbol{W}_K^i, \boldsymbol{W}_Q^i \in \mathbb{R}^{m \times d}$, $\boldsymbol{W}_2 \in \mathbb{R}^{d \times r}$, and $\boldsymbol{W}_1 \in \mathbb{R}^{r \times d}$.

We denote $t^{h,m,r} : \mathbb{R}^{d \times T} \to \mathbb{R}^{d \times T}$ as a transformer block with an attention layer with $h$ heads of size $m$, and a feed-forward layer with $r$ hidden nodes. Thus, the transformer can be written as:

$$\mathcal{T}^{h,m,r} := \left\{ f : \mathbb{R}^{d \times T} \to \mathbb{R}^{d \times T} \mid f \text{ is a composition of transformer blocks } t^{h,m,r} \right\}. \tag{7}$$

Similarly, transformer with absolute positional embedding is:

$$\mathcal{T}_{\text{P}}^{h,m,r} := \left\{ f_{\text{P}}(\boldsymbol{X}) = f(\boldsymbol{X} + \boldsymbol{E}) \mid f \in \mathcal{T}^{h,m,r} \text{ and } \boldsymbol{E} \in \mathbb{R}^{d \times T} \right\} \tag{8}$$

**Universal Approximator (UA)** The universal approximation framework considers the feasibility or existence of a neural network that can approximate different types of functions with arbitrarily small error. Consider a transformer network $f_1$ and an arbitrary function $f_2$, where $f_1, f_2 : \mathbb{R}^{n \times T} \to \mathbb{R}^{n \times T}$ are both sequence-to-sequence functions. We define a distance between $f_1$ and $f_2$ as:

$$\text{d}_p (f_1, f_2) := \left( \int \|f_1(\boldsymbol{X}) - f_2(\boldsymbol{X})\|_p^p \, d\boldsymbol{X} \right)^{1/p} \tag{9}$$

being a UA means that for any given $f_2 \in \mathcal{F}$, let $1 \le p < \infty$ and $\epsilon > 0$, there exists a network $f_1$, such that $\text{d}_p(f_1, f_2) \le \epsilon$. Several prior works have explored the concept of universal approximators (UAs) and whether transformers qualify as UAs. Below, we outline the key results from the literature that will be referenced in this paper:

**Theorem 2** (informal, see Yun et al. (2019)). Given $1 \le p < \infty$ and $\epsilon > 0$, for any function $f \in \mathcal{F}_{\text{PE}}$, where $\mathcal{F}_{\text{PE}}$ consists of all continuous permutation equivariant functions with compact support, there exists a Transformer network $f \in \mathcal{T}^{2,1,4}$ where $\text{d}_p(f, g) \le \epsilon$.

**Theorem 3** (informal, see Yun et al. (2019)). Given $1 \le p < \infty$ and $\epsilon > 0$, for any function $f \in \mathcal{F}_{\text{CD}}$, where $\mathcal{F}_{\text{CD}}$ consists of all continuous functions with compact support, there exists a Transformer network $f \in \mathcal{T}_{\text{P}}^{2,1,4}$ where $\text{d}_p(f, g) \le \epsilon$.

Theorem 2 discussed that transformers without positional embeddings are UAs for all continuous permutation equivariant functions; and Theorem 3 discussed that transformers with absolute positional embeddings (APE) are UAs for all continuous functions with compact support. Note that the latter results may be overruled by modifying the transformer architectures as follows:

**Theorem 4** (informal, see Luo et al. (2022)). Consider relative positional encoding (RPE) that modifies the attention as $\text{Attn}(\boldsymbol{X}) = \boldsymbol{X} + \sum_{i=1}^{h} \boldsymbol{W}_O^i \boldsymbol{W}_V^i \boldsymbol{X} \cdot \text{softmax} \left[ \left( \boldsymbol{W}_K^i \boldsymbol{X} \right)^T \boldsymbol{W}_Q^i \boldsymbol{X} + \mathbf{E} \right]$, where $\mathbf{E} \in \mathbb{R}^{T \times T}$ encodes the relative position within attention maps. Given $T > 2$, there always exists a continuous function $f_M : \mathcal{D} \to \mathbb{R}^{d \times T}$, such that $\sup_{\boldsymbol{X} \in \mathcal{D}} \|f_{\text{RPE}}(\boldsymbol{X}) - f_M(\boldsymbol{X})\|_2^2 \ge M$ holds for any modified RPE-based transformer.

While UA framework is typically used for understanding the approximation problem towards continuous functions, more recently, it is used to understand the approximation problem towards discontinuous functions. Specifically, in this work, we reference the results in Ismailov (2023) that

shows any discontinuous function may be implemented by a three-layer Kolmogorov type neural network:

**Theorem 5** (informal, see Ismailov (2023)). Given $d \geq 2$ and any function $f : \mathbb{I}^d \to \mathbb{R}$, where $\mathbb{I}$ is a closed unit internal $[0, 1]$, then function $f$ can be implemented exactly by a three-layer Kolmogorov neural network with $d$, $2d+1$, and $1$ processing units in the first, second, and final layer, respectively.

As stated in Ismayilova & Ismailov (2023); Ismailov (2023), the expressiveness of simple neural networks can be extended by constructing more diverse activation functions, which helps us understand the approximation ability towards discontinuous functions which are more prevalent in real-world complex systems. Note that other works also discuss the approximation problems towards functions that may be discontinuous (Kidger & Lyons, 2020; Pinkus, 1999).

## A.2 ADDITIONAL PROOF OF THEOREM 1

We only study the convergence property of self attention layers as the convergence property of feed forward networks has been extensively studied in previous works. To prove convergence, we build an input sequence $\mathbf{X} + \Delta_n$, where $\Delta_n$ is defined as a bounded perturbation matrix $\Delta \in \mathcal{D}$ that is uniformly scaled by a positive value $n$. Given a self attention layer $\text{Attn}(\mathbf{X})$, we show the following:

**Lemma 1.** Given $n \geq N$, $\mathbf{X} \in \mathcal{D}$, $\Delta_n = \Delta/n$, there exists an $\epsilon$ such that:

$$\sup_{\Delta \in \mathcal{D}, \|\Delta\|_1 \leq 1} \| \text{Attn}(\mathbf{X}) - \text{Attn}(\mathbf{X} + \Delta_n) \|_2 < \epsilon. \tag{10}$$

holds for any self attention layer parameterized by $\boldsymbol{W}_O^i \in \mathbb{R}^{d \times m}$, $\boldsymbol{W}_V^i, \boldsymbol{W}_K^i, \boldsymbol{W}_Q^i \in \mathbb{R}^{m \times d}$.

**Proof.** First, we re-write the activation component $\sigma \left[ \left( \boldsymbol{W}_K^i \boldsymbol{X} \right)^\top \boldsymbol{W}_Q^i \boldsymbol{X} \right]$ in Eq. 6 as column-wise softmax operation $\text{softmax}(\mathbf{X}^\top \mathbf{W} \mathbf{x}_j)$, where $\mathbf{x}_j$ is a random column of $\mathbf{X}$. We have:

$$
\begin{aligned}
&\| \text{softmax}(\mathbf{X}^\top \mathbf{W} \mathbf{x}_j) - \text{softmax}((\mathbf{X} + \Delta_n)^\top \mathbf{W}(\mathbf{x}_j + \delta_{j,n})) \|_2 \\
&\leq \| \text{softmax}(\mathbf{X}^\top \mathbf{W} \mathbf{x}_j) - \text{softmax}((\mathbf{X} + \Delta_n)^\top \mathbf{W}(\mathbf{x}_j + \delta_{j,n})) \|_1 \\
&\leq 2 \| \mathbf{X}^\top \mathbf{W} \mathbf{x}_j - (\mathbf{X} + \Delta_n)^\top \mathbf{W}(\mathbf{x}_j + \delta_{j,n}) \|_\infty \text{ (Corollary A.7 in Edelman et al. (2022))} \\
&= 2 \max_i \left( \frac{1}{n}(\mathbf{x}_i^\top \mathbf{W} \delta_j + \delta_i^\top \mathbf{W} \mathbf{x}_j) + \frac{1}{n^2} \delta_i^\top \mathbf{W} \delta_j \right) \\
&\leq 2 \max_i \left( \frac{1}{n}(\mathbf{x}_i^\top \mathbf{W} \mathbb{1} + \mathbb{1}^\top \mathbf{W} \mathbf{x}_j) + \frac{1}{n^2} \mathbb{1}^\top \mathbf{W} \mathbb{1} \right) = \epsilon_h
\end{aligned}
\tag{11}
$$

which shows that the attention map converges given deterministic $\mathbf{X}$ and $\mathbf{W} = (\mathbf{W}_K^i)^\top \mathbf{W}_Q^i$. Thus, Eq. 10 holds by considering the self attention operator $\text{Attn}(\mathbf{X})$ as a convex combination of attention heads given deterministic $\mathbf{X}$, $\boldsymbol{W}_O^i \in \mathbb{R}^{d \times m}$, $\boldsymbol{W}_V^i \in \mathbb{R}^{m \times d}$.

Thus, given $\boldsymbol{X} \to \mathbf{0}$, the transformer network $f_P(\mathbf{X})$ converges to a deterministic matrix $\mathbf{B}$ that is dependent on the network parameters and the positional embedding $\mathbf{E}$. Note that, under the context of network optimization, a more generalized version of the convergence property of transformers has been proved in other previous works (Wu et al., 2024; Gao et al., 2024).

## A.3 ADDITIONAL PROOF OF PROPOSITION 1

Based on the results in Theorem 3, it is known that as long as the constructed sequence $\mathbf{X} = [\mathcal{E}(\mathbf{S}_1), \mathcal{E}(\mathbf{S}_2), ..., \mathcal{E}(\mathbf{S}_T)]$ forms a continuous sequence-to-sequence function between input $\mathbf{X}$ and the target $\{A[g(t_i)]\}_{i=1}^T$, it is guaranteed that there exists a transformer network $f_P \in \mathcal{T}_P^{2,1,4}$ that can approximate the constructed sequence-to-sequence function. Thus, we show how the presented two conditions are sufficient to meet the above requirement:

- When there exists a continuous mapping between a fixed element $p$ of $\mathbf{S}_i$ and the $i$-th element of the target output $A[g(t_i)]$, one can construct a simple linear encoder $\mathcal{E}(\mathbf{S}) = \mathbf{S}\mathbf{v}$, where $\mathbf{v}[i] = 0$ when $i \neq p$ and $\mathbf{v}[p] = 1$, that creates the continuous sequence-to-sequence function.
- Based on the results in Theorem 5, if there exists an expressive tokenizer $\mathcal{E}$ (that may be a discontinuous function) that preprocess $\mathbf{S}_i$ to create a continuous mapping between $\mathcal{E}(\mathbf{S}_i)$ to the target, the existence of the transformer is guaranteed for a continuous sequence-to-sequence function.

**Example solutions** We provide example solutions to approximate the differential operator. Under the first condition, a trivial solution can be constructed by performing phase transition with degradation operator, creating $(d_i \circ g)(t) = \sin(Mt + \frac{i\pi}{2T})/M$. While the first condition requires data-specific degradation operators, the second condition provides more flexibility. In the case of differential operator, we rely on the result from Ismailov (2023) and use a Kolmogorov's mapping three-layer neural network to approximate an arbitrary (continuous or discontinuous) function $f : \mathbb{I}^T \to \mathbb{R}$, where $\mathbb{I}$ is a compact interval $[0, 1]$. Thus, one can construct a simple degradation operator of value shifts as: $(d_i \circ g)(t) = \sin(Mt)/M + (1 + i\delta/T)/M$, where $\delta \in (0, M-2]$ is an arbitrary number that distinguishes $\mathbf{S}_i$. In this case, there exists an encoder $\mathcal{E}$ that can create a continuous sequence-to-sequence function to the desired target (e.g., $\mathcal{E}(\mathbf{S}_i) = Mt_i$), where the existence of a solution is guaranteed by previous results in Yun et al. (2019).

# B EXPERIMENTAL DETAILS

## B.1 SYNTHETIC EXPERIMENTS

### B.1.1 SYNTHETIC DATASETS DETAILS

**Fractional Brownian motion (fBm)** Given a Hurst index $\mathcal{H}$ and two time steps $i$ and $j$, a fBm process is a continuous-time Gaussian process $B_{\mathcal{H}}(t)$ with the following covariance structure:

$$E\left[B_{\mathcal{H}}(i)B_{\mathcal{H}}(j)\right] = \frac{1}{2}\left(|i|^{2\mathcal{H}} + |j|^{2\mathcal{H}} - |i-j|^{2\mathcal{H}}\right) \tag{12}$$

Define function $\gamma(i, \mathcal{H}) = 0.5(|i-1|^{2\mathcal{H}} + |i+1|^{2\mathcal{H}} - 2|i|^{2\mathcal{H}})$, a fBm process can be simulated through the Cholesky decomposition method detailed as follows:

---

**Algorithm 1** Simulation of fBm processes using the Cholesky's method

---

**Inputs:** $N$ as the length of sequence (time steps), $\mathcal{H} \in (0, 1)$ as the Hurst index
**Initialize:** $\mathbf{L} \in \mathbb{R}^{N \times N}$, $\mathbf{V} \in \mathbb{R}^N$ with each entry randomly sampled from $\mathcal{N}(0, 1)$
**Define:** $\mathbf{X} \in \mathbb{R}^N$ as the output vector
    Initial conditions for $\mathbf{L}$: $\mathbf{L}[0, 0] = 1$, $\mathbf{L}[1, 0] = 2^{2\mathcal{H}-1} - 1$, $\mathbf{L}[1, 1] = (1 - \mathbf{L}[1, 0]^2)^{1/2}$,
    Initial conditions for $\mathbf{X}$: $\mathbf{X}[0] = \mathbf{V}[0]$, $\mathbf{X}[1] = \mathbf{L}[1, 0]\mathbf{V}[0] + \mathbf{L}[1, 1]\mathbf{V}[1]$
**for** each time step $i$ from 2 till $N - 1$ **do** $\mathbf{L}[i, 0] = \gamma(i, \mathcal{H})$
    **for** each time step $j$ from 1 to $i - 1$ **do**

$$\mathbf{L}[i, j] = \frac{1}{\mathbf{L}[j, j]}\left(\gamma(i - j, \mathcal{H}) - \sum_{k=0}^{j-1} \mathbf{L}[i, k] \cdot \mathbf{L}[j, k]\right)$$

    **end for**
    Update $\mathbf{L}[i, i] = (1 - \sum_{k=0}^{i-1}(\mathbf{L}[i, k]^2))^{1/2}$, $X[i] = \sum_{k=0}^{i} \mathbf{L}[i, k]\mathbf{V}[k]$
**end for**
**for** each time step $i$ from $N - 1$ till 0 **do** $\mathbf{X}[i] = (\sum_{k=0}^{i} \mathbf{X}[k]) \times N^{-\mathcal{H}}$
**end for**
**Output:** A simulated fBm process $\mathbf{X}$

---

**Autocorrelated sinusoids** The autocorrelated sinusoids dataset is generated with AR processes in the frequency space. Given an integer $k$, a randomly initialized set of weights $\{\phi_i\}_{i=1}^k$, an AR($k$) process defines the sequence of frequency values as follows:

$$f_t = \sum_{i=1}^{k} \phi_i f_{t-i} \tag{13}$$

The AR process ensures that the frequency components in the synthetic dataset are correlated, creating an autoregressive frequency structure that NoTS can effectively learn from.

### B.1.2 Feature regression task details

We detail the feature extraction methods as follows. We use them as the ground truth for the feature regression task. Define the indicator function $\mathbf{1}_A(x)$, where $\mathbf{1}_A(x) = 1$ if $x \in A$ and $\mathbf{1}_A(x) = 0$ otherwise. Given a single-channel signal $\mathbf{s} \in \mathbb{R}^T$ with $v_i$ as the value on $i$-th timestamp, all features are extracted on each channel of the signal as follows:

**Slope Sign Change (SSC)** SSC measures directional slope changes in a signal, indicating the intensity of fluctuations. Given a threshold value $\delta$ as hyperparameter, a period of time series sequence $\mathbf{s}$, we extract SSC value with the following equation:

$$\text{SSC}(\mathbf{s}) = \sum_{i=2}^{T-1} \mathbf{1}_{(v_i - v_{i-1})(v_i - v_{i+1}) < 0}(v_i) \cdot \mathbf{1}_{\max(|v_i - v_{i+1}|, |v_i - v_{i-1}|) \geq \delta}(v_i)$$

In practice, we extract the SSC values on top of segmented signals with a length of 32.

**Willison Amplitude (WAMP)** WAMP is a similar feature that focuses on counting significant amplitude changes between consecutive steps. Given a threshold value $\delta$ as hyperparameter, a period of time series sequence $\mathbf{s}$, WAMP is computed through $\text{WAMP}(\mathbf{s}) = \sum_{i=1}^{T-1} \mathbf{1}_{|v_{i+1} - v_i| \geq \delta}(v_i)$. In practice, we also extract the WAMP values on top of segmented signals with a length of 32, creating a 32-dimensional feature for each studied synthetic data sample.

**Band power (b. power)** The band power quantifies the energy within a specific selected range of frequencies. It is computed by first performing the Fourier transform of $\mathbf{s}$, creating a frequency representation $\mathbf{s}(f)$. The band power within frequency range $[f_1, f_2]$ is later extracted as $\text{BP}_{(f_1, f_2)}(\mathbf{s}) = \int_{f_1}^{f_2} |\mathbf{s}(f)|^2 \, df$. In this work, we consider 3 unique frequency range $\{[5, 10], [15, 30], [30, 80]\}$ as hyperparameters to extract a 96-dimensional feature for each studied synthetic data sample.

## B.2 Real-world experiments

### B.2.1 Dataset information

**Classification** We selected 9 univariate datasets from the UCR archive (Dau et al., 2019), filtering out all datasets with less than 140 series length or less than 350 training samples. The dataset selection is performed to ensure each dataset has both sufficient samples and dynamics. The detailed information about the selected datasets is provided in Table 4.

| Dataset | Train | Test | Series Length | Classes |
|---|---|---|---|---|
| FordA | 3601 | 1320 | 500 | 2 |
| FordB | 3636 | 810 | 500 | 2 |
| ScreenType | 375 | 375 | 720 | 3 |
| ECG5000 | 500 | 4500 | 140 | 5 |
| Wafer | 1000 | 6164 | 152 | 2 |
| StarLightCurves | 1000 | 8236 | 1024 | 3 |
| UWaveGestureLibraryAll | 896 | 3582 | 945 | 8 |
| HandOutlines | 1000 | 370 | 2709 | 2 |
| EthanolLevel | 504 | 500 | 1751 | 4 |

Table 4: Detailed information about the selected datasets from the UCR archive.

We also selected 5 multivariate datasets from the UEA archive (Bagnall et al., 2018), excluding those with a series length below 100 and the training sample size below 200. The detailed information about the selected datasets is provided in Table 5.

| Dataset | Channel | Train | Test | Series Length | Classes |
|---|---|---|---|---|---|
| EthanolConcentration | 3 | 261 | 263 | 1751 | 4 |
| Heartbeat | 61 | 204 | 205 | 405 | 2 |
| PEMS-SF | 963 | 267 | 173 | 144 | 7 |
| SelfRegulationSCP1 | 6 | 268 | 293 | 896 | 2 |
| SelfRegulationSCP2 | 7 | 200 | 180 | 1152 | 2 |

Table 5: Detailed information about the selected datasets from the UEA archive.

**Imputation**  For the imputation tasks, we use the ETDataset (Zhou et al., 2021), where ETTm1 and ETTm2 are sampled at minute intervals, and ETTh1 and ETTh2 are sampled at hourly intervals. The detailed information about the selected datasets is provided in Table 6.

| Dataset | Channel | Series Length | Train | Validation | Test |
|---|---|---|---|---|---|
| ETTm1, ETTm2 | 7 | 96 | 34465 | 11521 | 11521 |
| ETTh1, ETTh2 | 7 | 96 | 8545 | 2881 | 2881 |

Table 6: ETDataset for imputation tasks.

**Anomaly detection**  The detailed information about the selected datasets is provided in Table 7.

| Dataset | Channel | Series Length | Train | Validation | Test |
|---|---|---|---|---|---|
| SMD | 38 | 100 | 566724 | 141681 | 708420 |
| MSL | 55 | 100 | 44653 | 11664 | 73729 |
| SWaT | 51 | 100 | 396000 | 99000 | 449919 |
| PSM | 25 | 100 | 105984 | 26497 | 87841 |

Table 7: Detailed information about the selected datasets for the anomaly detection tasks.

### B.2.2 MODEL TRAINING AND ARCHITECTURE DETAILS

**Training details**  For pre-training on synthetic datasets, we use a learning rate of $0.05$, a Multi-StepLR scheduler with a multiplicative factor $\gamma = 0.3$, and two milestones on epoch 30 and 150. We perform all pre-training for a total of 300 epochs on both of the synthetic datasets, where we set batch size as 1024 for the reconstruction task.

For pre-training on real-world datasets, we use a learning rate of $0.005$. We perform all pre-training for either a total of 300 epochs, or a total of 6000 steps, whichever finishes the first. We set batch size as 32 for imputation and anomaly detection tasks, and batch size as 64 for classification tasks.

Table 8: Complete classification results on the UCR datasets.

| Dataset | Parameter efficient tuning | | | | Full-scale fine-tuning | | | |
|---|---|---|---|---|---|---|---|---|
| | NoTS-lw | Next-pred | bioFAME | SimMTM | NoTS-lw | Next-pred | bioFAME | SimMTM |
| HandOutlines | 71.62 | 64.32 | 64.05 | 88.92 | 93.51 | 72.16 | 91.62 | 89.73 |
| EthanolLevel | 28.60 | 26.60 | 25.20 | 29.00 | 91.40 | 48.60 | 41.20 | 38.00 |
| StarLightCurves | 91.66 | 87.34 | 85.15 | 88.59 | 97.21 | 97.49 | 97.56 | 97.39 |
| UWave-GL-All | 67.39 | 56.28 | 37.38 | 76.35 | 96.57 | 96.90 | 87.83 | 94.72 |
| FordA | 81.52 | 77.27 | 71.74 | 51.59 | 94.02 | 94.09 | 93.49 | 94.55 |
| FordB | 68.27 | 63.83 | 57.41 | 64.57 | 83.70 | 86.17 | 85.19 | 83.58 |
| Wafer | 98.78 | 89.21 | 89.21 | 89.23 | 99.81 | 99.87 | 99.64 | 99.85 |
| ECG5000 | 91.31 | 89.00 | 92.96 | 87.38 | 94.13 | 88.47 | 94.29 | 93.82 |
| ScreenType | 47.73 | 39.73 | 40.53 | 42.67 | 42.40 | 41.87 | 42.93 | 43.20 |
| **Average** | 71.88 | 65.95 | 62.63 | 68.70 | 88.08 | 80.62 | 81.53 | 81.65 |

| | **Attaching NoTS to existing architectures** | | | |
|---|---|---|---|---|
| **Dataset** | PatchTST | PatchTST + **NoTS** | iTransformer | iTransformer + **NoTS** |
| HandOutlines | 91.89 | 93.51 | 92.16 | 92.16 |
| EthanolLevel | 57.80 | 68.00 | 86.20 | 86.40 |
| StarLightCurves | 97.46 | 97.56 | 93.94 | 93.52 |
| UWave-GL-All | 96.04 | 96.45 | 89.89 | 91.76 |
| FordA | 93.71 | 93.56 | 77.05 | 83.11 |
| FordB | 78.64 | 80.49 | 68.52 | 69.14 |
| Wafer | 99.59 | 99.63 | 99.72 | 99.77 |
| ECG5000 | 94.09 | 94.33 | 94.42 | 94.49 |
| ScreenType | 42.93 | 44.00 | 42.13 | 45.07 |
| **Average** | 83.57 | 85.28 | 82.67 | 83.94 |

We apply the same set of hyperparameters for both parameter efficient fine-tuning and full-scale fine-tuning, where we perform hyperparameter selection on learning rate $\{0.005, 0.001, 0.05\}$ and batch size $\{32, 64, 128\}$. We perform the fine-tuning for 300 epochs on imputation, anomaly detection, and feature regression tasks, and perform the fine-tuning for 4000 steps on classification tasks.

The settings are applied consistently across all models to ensure a fair comparison. All models are optimized with an Adam optimizer with $\beta_1 = 0.9$, $\beta_2 = 0.99$, and a weight decay of $1 \times 10^{-5}$.

**Model architectures** For all pre-training methods including NoTS-lw, we use a same channel-independent 1D-ResNet encoder for fair comparison. The encoder has 3 ResNet layers of channel size $\{16, 32, 64\}$, each has 2 ResNet blocks. The first convolutional layer has a kernel size of 7, and the rest layers have a kernel size of 3. We append an additional convolutional layer after the ResNet blocks to alter the dimensionality $d$ of the token embeddings, where model variant $d = 32$ is used for all experiments, and $d = 16, 64, 128$ is trained for the scalability pilot study. We use a 3-layer 4-head transformer with a token dimension of $d$, and $4\times$ size in the feed forward layer. The decoder is built to be symmetric to the encoder architecture.

Table 9: Complete classification results on the UEA datasets.

| | **Parameter efficient tuning** | | | | **Full-scale fine-tuning** | | | |
|---|---|---|---|---|---|---|---|---|
| **Dataset** | **NoTS-lw** | Next-pred | bioFAME | SimMTM | **NoTS-lw** | Next-pred | bioFAME | SimMTM |
| EthanolConcentration | 28.14 | 25.48 | 28.14 | 25.48 | 30.04 | 29.28 | 27.76 | 28.90 |
| Heartbeat | 73.66 | 73.17 | 72.68 | 72.20 | 74.63 | 73.66 | 73.17 | 73.17 |
| PEMS-SF | 75.72 | 58.38 | 77.46 | 64.16 | 80.35 | 67.63 | 75.15 | 72.25 |
| SelfRegulationSCP1 | 79.18 | 77.82 | 69.97 | 59.39 | 89.08 | 86.01 | 85.67 | 74.06 |
| SelfRegulationSCP2 | 57.22 | 56.67 | 53.33 | 55.56 | 57.78 | 57.22 | 56.11 | 57.78 |
| **Average** | 62.78 | 65.95 | 62.63 | 68.70 | 88.08 | 80.62 | 81.53 | 81.65 |

| | **Attaching NoTS to existing architectures** | | | |
|---|---|---|---|---|
| **Dataset** | PatchTST | PatchTST + **NoTS** | iTransformer | iTransformer + **NoTS** |
| EthanolConcentration | 25.10 | 25.48 | 30.42 | 30.04 |
| Heartbeat | 73.17 | 74.63 | 73.17 | 74.15 |
| PEMS-SF | 88.44 | 90.75 | 89.02 | 90.75 |
| SelfRegulationSCP1 | 78.16 | 82.25 | 87.71 | 88.06 |
| SelfRegulationSCP2 | 51.67 | 51.67 | 57.78 | 58.33 |
| **Average** | 63.31 | 64.96 | 67.62 | 68.27 |

In our experiments, iTransformer (Liu et al., 2023b) is implemented to transform inputs to an embedding dimension of 16, where we build the decoder to be symmetric and linear. We used a 3-layer 4-head transformer network with a token dimensionality of 128 and $4\times$ size in the feed forward layer. PatchTST (Nie et al., 2022) is implemented with a patch length of 16, stride of 8, and token dimension of 32. In cases where PatchTST becomes too computationally heavy (e.g., anomaly detection tasks), we adjust batch size to be 1 and increase patch length to be 32.

Table 10: Complete imputation results with masking ratio 12.5% and 25%.

| Dataset | Parameter efficient tuning | | | | Full-scale fine-tuning | | | |
|---|---|---|---|---|---|---|---|---|
| | NoTS-lw | Next-pred | bioFAME | SimMTM | NoTS-lw | Next-pred | bioFAME | SimMTM |
| *12.5% masking ratio* | | | | | | | | |
| ETTm1 | 0.1556 | 0.2832 | 0.1957 | 0.1573 | 0.1194 | 0.1219 | 0.1251 | 0.1207 |
| ETTm2 | 0.1232 | 0.1774 | 0.1183 | 0.1243 | 0.1110 | 0.1164 | 0.1038 | 0.1041 |
| ETTh1 | 0.2764 | 0.4569 | 0.2471 | 0.2545 | 0.2091 | 0.2126 | 0.1966 | 0.1947 |
| ETTh2 | 0.1917 | 0.2692 | 0.1746 | 0.1796 | 0.1615 | 0.1886 | 0.1751 | 0.1632 |
| *25% masking ratio* | | | | | | | | |
| ETTm1 | 0.1730 | 0.3280 | 0.2103 | 0.1697 | 0.1246 | 0.1377 | 0.1325 | 0.1244 |
| ETTm2 | 0.1294 | 0.1789 | 0.1257 | 0.1269 | 0.1205 | 0.1218 | 0.1095 | 0.1093 |
| ETTh1 | 0.2957 | 0.4738 | 0.2695 | 0.2734 | 0.2266 | 0.2440 | 0.2068 | 0.2078 |
| ETTh2 | 0.1994 | 0.2707 | 0.1824 | 0.1861 | 0.1653 | 0.1872 | 0.1806 | 0.1681 |

| Dataset | Attaching NoTS to existing architectures | | | |
|---|---|---|---|---|
| | PatchTST | PatchTST + **NoTS** | iTransformer | iTransformer + **NoTS** |
| *12.5% masking ratio* | | | | |
| ETTm1 | 0.1791 | 0.1657 | 0.1539 | 0.1662 |
| ETTm2 | 0.1233 | 0.1193 | 0.1082 | 0.1071 |
| ETTh1 | 0.3277 | 0.2705 | 0.2325 | 0.2227 |
| ETTh2 | 0.1817 | 0.1797 | 0.1639 | 0.1609 |
| *25% masking ratio* | | | | |
| ETTm1 | 0.1837 | 0.1903 | 0.1698 | 0.1665 |
| ETTm2 | 0.1295 | 0.1268 | 0.1140 | 0.1117 |
| ETTh1 | 0.3668 | 0.2952 | 0.2483 | 0.2318 |
| ETTh2 | 0.1926 | 0.1827 | 0.1725 | 0.1678 |

### B.2.3 COMPLETE EXPERIMENTAL RESULTS

We show the complete classification results in Table 8 and Table 9 and the complete imputation results in Table 10. The averaged results are presented in Table 2.

**Imputation task details** We perform a channel-wise imputation task instead of the traditional random imputation task. Specifically, when performing the masking, instead of uniformly sample random elements from $C \times T$ entries of $\mathbf{S} \in \mathbb{R}^{C \times T}$, we sample uniformly from $T$ columns and cover inputs from all channels. This is to eliminate the overfitting issues from the data embeddings.

### B.2.4 ADDITIONAL VISUALIZATIONS

We present additional data and token space visualizations in Figure 4 and Figure 5, respectively.

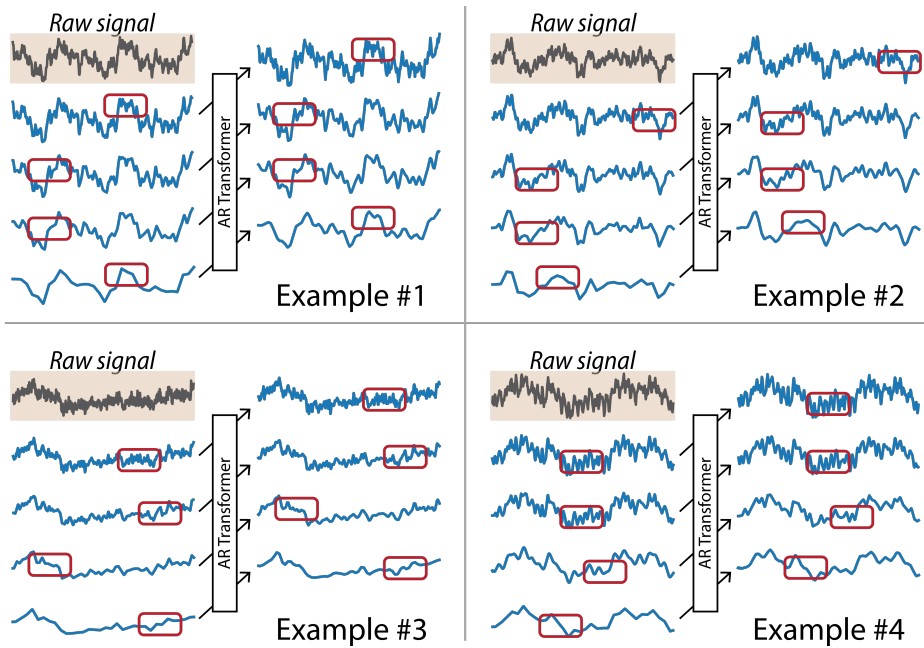

Figure 4: Additional data space visualizations.

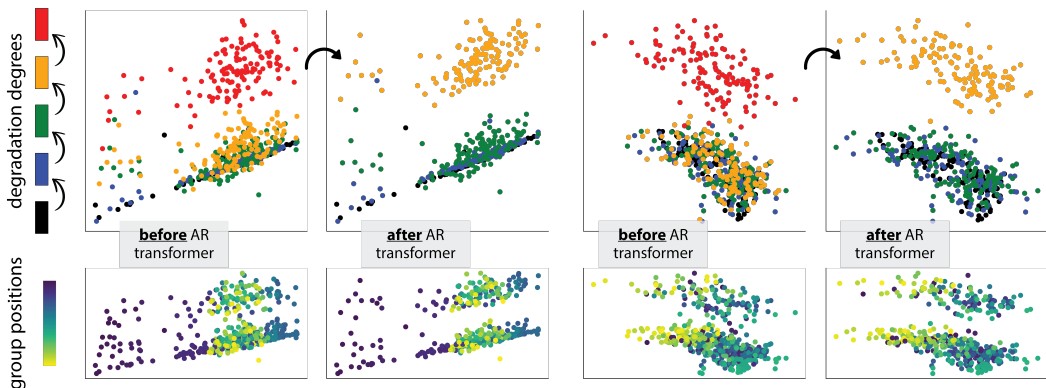

Figure 5: Additional token space visualizations.

## C DETAILED MODEL DEPLOYMENT PIPELINE

### C.1 OVERALL PRE-TRAINING, FINE-TUNING, AND TESTING PIPELINES

#### C.1.1 OVERALL MODEL DEPLOYMENT FLOW

The proposed NoTS method, as a pre-training strategy, considers the following model training and deployment flow. It considers a pre-training datasets $D_{PT}$ and $D_{FT}$ with samples shaped $C_{PT} \times T_{PT}$ and $C_{FT} \times T_{FT}$, respectively, allowing for differing channel and temporal dimensions between these two phases. The model is first pre-trained by performing the autoregressive reconstruction task, guided by the training objective as detailed in Equation 3. It is later fine-tuned on the training split of the downstream dataset, and is finally evaluated on the testing split of the downstream dataset. Depending on the differences between pre-training and downstream datasets, there are two schemes as presented in Table 11.

During pre-training, the input time series are sequentially processed by the encoding layers, the autoregressive transformer, and the decoding layers for the reconstruction task. In fine-tuning, ran-

| Stage | Training | Tuning | Testing |
|---|---|---|---|
| Cross-domain | Synthetic | Real-world (train split) | Real-world (test split) |
| In-domain | Real-world (train split) | Real-world (train split) | Real-world (test split) |

Table 11: Data splits for training, tuning, and testing.

domly initialized channel adapters and prompt task adapters are added, either trained independently or jointly with the pre-trained encoder, decoder, and transformer during the prompt tuning and full-scale fine-tuning stages, respectively. Finally, the tuned model, with all parameters frozen, is used for testing. The process is detailed as in Table 12.

| Stage | Training | Tuning | Testing |
|---|---|---|---|
| Prompt tuning | $\mathcal{E}, \mathcal{D}$, Transformer | Adaptors | NA |
| Full-scale tuning | $\mathcal{E}, \mathcal{D}$, Transformer | $\mathcal{E}, \mathcal{D}$, Transformer, Adaptors | NA |

Table 12: Parameters updated during training, tuning, and testing.

### C.1.2 DOMAIN SHIFTS ACROSS PRE-TRAINING AND FINE-TUNING

Domain shift issues between pre-training and downstream datasets are discussed in many recent works on pre-training methods for time series data (Dong et al., 2024; Liu et al., 2023a). These challenges primarily stem from differences between $D_{PT}$ and $D_{FT}$, leading to the following two cross-domain and within-domain schemes:

**Cross-domain pre-training and fine-tuning.** All NoTS-lw models are trained and evaluated using the cross-domain scheme (first 8 rows of Table 2). Specifically, the models are pre-trained on the synthetic datasets detailed in Section 5.1 to learn and extract the universal dynamic features, then tuned on real-world datasets to demonstrate the generalization capability of the pre-training approach.

**Within-domain pre-training and fine-tuning.** All NoTS models are trained and evaluated using the within-domain scheme (Table 1 and last 4 rows of Table 2). This setting assesses the model's ability to learn dataset-specific patterns by pre-training and fine-tuning on the same train split of a dataset. The final evaluation of the downstream task in fine-tuning is evaluated on the test split of the corresponding dataset.

### C.1.3 PROMPT TUNING AND FULL-SCALE FINE-TUNING

We design the prompt tuning method to adapt pre-trained knowledge to new tasks while introducing minimal additional parameters. In this tuning paradigm, the pre-trained transformer (along with the encoder and decoder) remains frozen. Only the adaptor parameters (channel and task adaptors) and, for discriminative tasks, a final linear classification head are trained.

In contrast, the full-scale fine-tuning method unfreezes the pre-trained weights, allowing the entire model, including the encoder, transformer layers, the decoder, the adaptors, and the final linear classification head, to be fine-tuned. This approach enables comprehensive adaptation to new data and tasks but comes with higher computational demands.

### C.1.4 TESTING SCHEMES

The model is tested on the testing split of the downstream dataset. The specific testing scheme varies across different downstream tasks, with major differences across generative (anomaly detection and imputation tasks) and discriminative tasks (classification and regression tasks). More details are included in the downstream tasks section.

## C.2 MATHEMATICAL FORMULATION OF OUR PROMPT-TUNING STRATEGY

### C.2.1 TASK ADAPTORS INSPIRED BY VISUAL PROMPT TUNING (VPT)

Consider the input sequence to transformer layer $\mathbf{X} = [\mathbf{x}_1, \mathbf{x}_2, \ldots, \mathbf{x}_N] \in \mathbb{R}^{N \times D}$, where $N$ is the sequence length and $D$ is the transformer's hidden dimension. Our prompt tuning pipeline deploys the deep visual prompt tuning strategy as follows:

We randomly initialize a set of $k$ learnable prompt tokens $\{p_1, \ldots, p_k\}$, where each $p_i \in \mathbb{R}^D$. For each transformer layer $d$, suppose we have $H$ attention heads, and we initialize three separate linear projection layers for the prompt tokens:

$$L_{q_p}^{(d)}, \quad L_{k_p}^{(d)}, \quad L_{v_p}^{(d)} \in \mathbb{R}^{D \times (H \times D)},$$

each without bias. These layers project the prompt tokens into query, key, and value representations of shape $\mathbb{R}^{H \times D}$ as follows:

$$p_{q_i}^{(d)} = p_i L_{q_p}^{(d)}, \quad p_{k_i}^{(d)} = p_i L_{k_p}^{(d)}, \quad p_{v_i}^{(d)} = p_i L_{v_p}^{(d)}, \quad \text{for } i = 1, 2, \ldots, k.$$

While the input tokens $\mathbf{x}_j \in \mathbb{R}^D$, where $j = 1, 2, \ldots, N$ have separate linear projection layers that are inherited from the pre-trained transformer attention blocks, giving $\mathbf{x}_{q_j}^{(d)}$, $\mathbf{x}_{k_j}^{(d)}$, and $\mathbf{x}_{v_j}^{(d)}$ for each $j$ in each transformer layer $d$. The projected prompt tokens and input tokens are then concatenated separately to form the augmented query, key, and value sequences:

$$\mathbf{Q}_{\text{aug}}^{(d)} = [p_{q_1}^{(d)}, p_{q_2}^{(d)}, \ldots, p_{q_k}^{(d)}, \mathbf{x}_{q_1}^{(d)}, \mathbf{x}_{q_2}^{(d)}, \ldots, \mathbf{x}_{q_N}^{(d)}],$$

$$\mathbf{K}_{\text{aug}}^{(d)} = [p_{k_1}^{(d)}, p_{k_2}^{(d)}, \ldots, p_{k_k}^{(d)}, \mathbf{x}_{k_1}^{(d)}, \mathbf{x}_{k_2}^{(d)}, \ldots, \mathbf{x}_{k_N}^{(d)}],$$

$$\mathbf{V}_{\text{aug}}^{(d)} = [p_{v_1}^{(d)}, p_{v_2}^{(d)}, \ldots, p_{v_k}^{(d)}, \mathbf{x}_{v_1}^{(d)}, \mathbf{x}_{v_2}^{(d)}, \ldots, \mathbf{x}_{v_N}^{(d)}].$$

These augmented query, key, and value sequences are then processed by the multi-head attention mechanism with $H$ heads in transformer layer $d$, allowing the model to integrate information from both the prompt tokens and the input time series tokens.

### C.2.2 CHANNEL ADAPTORS AND CHANNEL EMBEDDING

The channel embedding layer is applied before feeding data into the encoding layers, transforming the channel dimension from $C$ to $C'$ as follows:

$$L(\mathbf{S}) = W_{\text{embed}} \mathbf{X} + \mathbf{b}_{\text{embed}},$$

where $W_{\text{embed}} \in \mathbb{R}^{C' \times C}$ and $\mathbf{b}_{\text{embed}} \in \mathbb{R}^{C'}$ are learnable parameters.

Additionally, the learnable channel tokens $\mathbf{T} \in \mathbb{R}^{C \times D}$ are used as additive embeddings before the input sequence enters the transformer layers. These tokens recalibrate channel-specific representations, ensuring consistent adjustments across all tokens. This method enhances robustness and flexibility across diverse domains and channel configurations without requiring significant architectural changes.

## C.3 ADDITIONAL EXPERIMENTAL DETAILS FOR DOWNSTREAM TASKS

We follow the multitask evaluation pipeline in (Wu et al., 2022) and provide the necessary details as follows.

**Classification**  We consider the sequence-level classification task, where each time series sample is mapped into one label. The classification is performed by injecting prompt tokens into the task adapter, where the transformer processes the appended token sequence $[p_1, p_2, \ldots, p_k, x_1, x_2, \ldots, x_N] \in \mathbb{R}^{(N+k) \times D}$, where $k$ is the number of prompt tokens, $N$ is the number of input tokens, and $D$ is the transformer hidden dimension. We selected $k = 5$ for all classification tasks, where the prompt tokens are averaged and passed into a linear projection head to obtain classification logits.

**Anomaly detection** We consider the unsupervised time series anomaly detection task. During the training stage, we minimize the reconstruction error of the normal data using MAE loss. In testing, anomalies are detected by comparing reconstruction errors to a threshold derived from the error distribution of the entire dataset. We employ event-based anomaly detection and compute the detection accuracy for model evaluation. For all experiments, we sweep the anomaly rates across the data distribution {0.5%, 0.75%, 1.0%, 1.25%, 1.5%} to compute the final rate for each dataset.

**Imputation** We randomly generate and apply masks to the input to simulate and reconstruct the missing values. We evaluate imputation masking rates of 12.5% and 25%, with the mask ratio set to 1.5 times higher during training. The training objective minimizes the Mean Squared Error (MSE) between the ground truth and the model's imputed outputs, focusing on both masked points and the entire series.

**Regression** The regression task is performed exclusively on the synthetic dataset in a within-domain manner using a regression task adaptor during fine-tuning. The procedure is similar to the classification task, with the key difference being the loss function: classification uses cross-entropy loss, while regression employs MAE loss for fine-tuning.

## D    ADDITIONAL ABLATION RESULTS ON THE DEGRADATION OPERATORS

### D.1    PROPORTION OF GLOBAL AND LOCAL DEGRADATION OPERATORS

Table 13: Ablation study on the proportion of global and local degradation operators

| Global/Local | 0/100 | 25/75 | 50/50 | 75/25 | 100/0 |
|---|---|---|---|---|---|
| HandOutlines | 88.11 | 72.16 | 71.62 | 71.62 | 64.32 |
| EthanolLevel | 29.00 | 28.60 | 28.60 | 28.60 | 26.60 |
| StarLightCurves | 88.59 | 89.84 | 91.66 | 91.66 | 89.84 |
| UWave-GL-All | 65.32 | 64.15 | 67.39 | 57.87 | 56.28 |
| FordA | 80.22 | 81.89 | 81.52 | 87.20 | 80.38 |
| FordB | 64.57 | 68.52 | 68.27 | 70.00 | 74.07 |
| Wafer | 94.11 | 99.61 | 98.78 | 98.83 | 96.38 |
| ECG5000 | 91.31 | 92.96 | 91.31 | 91.31 | 94.09 |
| ScreenType | 43.19 | 45.07 | 47.73 | 44.00 | 44.00 |
| **Classification AVG** | 71.60 | 71.42 | 71.88 | 71.23 | 69.55 |

We conducted an ablation study to further assess the impact of varying the proportions of global and local operators on classification performance. Utilizing the prompt fine-tuning strategy outlined in our paper, we pre-trained four new variants of the NoTS model with different global/local operator ratios: 25/75, 75/25, and configurations where either global (0/100) or local (100/0) operators were entirely removed. These models were benchmarked against the original 50/50 proportion using classification tasks across nine datasets from the UCR repository. The results, summarized in Table 13, indicate that NoTS maintains robust performance across varying operator proportions. Notably, the optimal balance of global and local operators appears to be dataset-dependent. For instance, the HandOutlines dataset benefits more from a higher proportion of local degradation, whereas the FordB dataset performs better with increased global degradation. This suggests that NoTS can be effectively utilized to explore and adapt to the specific properties of different datasets, enhancing its versatility and applicability.

### D.2    INTENSITY OF DEGRADATION

To further investigate the role of degradation intensity in constructing functional sequences, we conducted two additional experiments aimed at uniformly decreasing and increasing the degradation intensity during the pre-training phase by adjusting smoothing kernels. The original model employed local smoothing kernels with values $\{2, 4, 8, 16\}$ and global smoothing kernels with frequency cutoff components $\{L/4, L/8, L/16, L/32\}$. In our new experiments, we adjusted these values to create varying intensities: decreasing intensity involved using larger smoothing kernels while increasing

Table 14: Ablation study on the intensity of degradation by adjusting smoothing kernels

| Intensity of Degradation | Decrease Intensity | Normal Intensity | Increase Intensity |
|---|---|---|---|
| Global Kernels | $\{256, 128, 64, 32\}$ | $\{128, 64, 32, 16\}$ | $\{64, 32, 16, 8\}$ |
| Local Kernels | $\{2, 4, 6, 8\}$ | $\{2, 4, 8, 16\}$ | $\{4, 8, 16, 32\}$ |
| HandOutlines | 65.28 | 71.62 | 78.92 |
| EthanolLevel | 26.60 | 28.60 | 26.60 |
| StarLightCurves | 87.34 | 91.66 | 91.09 |
| UWave-GL-All | 60.04 | 67.39 | 68.53 |
| FordA | 77.83 | 81.52 | 84.38 |
| FordB | 68.07 | 68.27 | 71.28 |
| Wafer | 96.28 | 98.78 | 96.32 |
| ECG5000 | 91.31 | 91.31 | 92.09 |
| ScreenType | 45.18 | 47.73 | 44.00 |
| **Classification AVG** | 68.66 | 71.88 | 72.58 |

intensity used smaller kernels. The selected kernels for each intensity level are detailed in Table 14. After fine-tuning the newly pre-trained models on the same classification tasks, we observed that increasing the intensity of degradation operators consistently enhanced classification performance, as evidenced by higher average accuracy scores. Conversely, decreasing the intensity had a detrimental effect on performance. These findings underscore the importance of appropriate degradation intensity in improving the model's generalization capabilities.

## D.3 NUMBER OF DEGRADATION STEPS

Table 15: Ablation study on the number of degradation steps

| Degradation Steps | 3 | 5 | 7 |
|---|---|---|---|
| Global Kernels | $\{128, 32\}$ | $\{128, 64, 32, 16\}$ | $\{128, 64, 32, 16, 8, 4\}$ |
| Local Kernels | $\{2, 8\}$ | $\{2, 4, 8, 16\}$ | $\{2, 4, 8, 16, 32, 64\}$ |
| HandOutlines | 72.70 | 71.62 | 83.52 |
| EthanolLevel | 26.60 | 28.60 | 31.00 |
| StarLightCurves | 85.49 | 91.66 | 87.74 |
| UWave-GL-All | 45.13 | 67.39 | 70.28 |
| FordA | 77.48 | 81.52 | 82.59 |
| FordB | 68.07 | 68.27 | 72.07 |
| Wafer | 90.42 | 98.78 | 98.74 |
| ECG5000 | 88.47 | 91.31 | 91.31 |
| ScreenType | 44.00 | 47.73 | 45.85 |
| **Classification AVG** | 66.48 | 71.88 | 73.69 |

We explored the effect of varying the number of degradation steps in the functional sequence to better explore the degradation operator's construction. Initially, our approach utilized a fixed sequence length of five degradation steps, including the raw signal. To assess the impact of this parameter, we altered the sequence length to three and seven degradation steps, following the same training pipeline. The corresponding global and local smoothing kernels for each configuration are presented in Table 15. Upon evaluation using the classification tasks, we found that increasing the number of degradation steps led to improved classification performance, with the average accuracy rising from 66.48 to 73.69 as the sequence length increased. However, this enhancement comes at the cost of higher memory usage during the pre-training phase, as the memory requirements scale linearly with the number of degradation steps. Notably, the inference stage's memory costs remain unaffected by this change. These results highlight a trade-off between model performance and computational resources, suggesting that while more degradation steps can bolster accuracy, they also demand greater memory capacity during training.