# OpenReview forum: "Generalizable autoregressive modeling of time series through functional narratives"
_ICLR.cc/2025/Conference — Submitted to ICLR 2025_

### Official Review · Reviewer_3Cgn · 2024-11-01

**Soundness:** 2
**Presentation:** 2
**Contribution:** 2
**Rating:** 5
**Confidence:** 4

**Summary:**

This paper proposes NoTS, a novel autoregressive pre-training method for time series. NoTS reinterprets time series as sequences of temporal functions generated by applying degradation operators of varying intensity levels to the original time series. The authors propose a "next-function prediction" task analogous to the next-word prediction task in language modelling and demonstrate the superior performance of NoTS on synthetic and real-world datasets.

**Strengths:**

* The proposed approach of next function prediction for time series pretraining is novel and very interesting. The authors motivate their method very well. The authors bring a very different perspective for modelling time series.
* The implementation of their approach with degradation operators is also well-theoretically justified. The authors theoretically show that degradation operators avoid the problem of discontinuities when time series are broken up into functions.

**Weaknesses:**

* Clarity: I am unable to understand many parts of the paper due to unclear presentation; especially relating to the empirical evaluation of the method.

* Evaluation: The protocol used for evaluation in Sec 5.2 is very much unclear. There are several references to Wu et al (2022) but the authors should not expect a reader to look into another paper, especially for the protocol of experimental setup, and the paper should be self-contained. I don't understand the protocol used even after looking into the paper. Do you pretrain on another dataset? If so, which dataset?

* Insufficient details on the training time, cost etc: It would be important to add the cost of using the architecture, both during pretraining and finetuning. In terms of training time / inference time / number of parameters etc.

**Questions:**

* Line 104: "giving a consistent performance boost when performing dataset-specific pre-training" - I don't understand what is dataset-specific pretraining. I have only heard of pretraining on a large dataset, and then dataset specific adaptation. Please clarify.

* Line 123: "trained while maintaining 82% average performance" - average performance on what? is it the pretraining dataset? it should be made clear.

* Line 294: "In time series domain, it is common to encounter datasets with new channel graphs or new tasks at test time. To apply the pre-trained weights in such situations, we build two types of adaptors" - you should give evidence (cite a paper) or provide an example for the first sentence. I don't understand what is meant by "encountering new channels at test time".

* Line 303: "We initialize and append prompt tokens to the transformer architecture along with data tokens following the deep visual prompt tuning plan as detailed in Jia et al. (2022)". Again, the authors should not expect a reader to look into another paper for such a crucial aspect of the model. I did not understand this part; the authors must explain the mechanism better.

* How sample-efficient is your model (does it do well few-shot)? Experiments like these can potentially substantiate your contribution.

* Why not test on the forecasting task too, which is, equally or maybe even more important than classification/imputation tasks?

---

> ### Author Response · Authors · 2024-11-28
> **Response to 3Cgn #1**
>
> Thank you for your insightful questions and comments. Below are our responses:
>
> ## W1 - Clarity issues of the work.
> Thank you for pointing out the clarity/presentation issues of the work, which we greatly appreciate. **We have put a great effort addressing those issues in the revised manuscript by incorporating the following texts into our revised manuscript**. This includes:
> - In Appendix, we add
> 1. A more comprehensive description and explanations of the model deployment technical details;
> 2. Mathematical formulation of our prompt-tuning strategy;
> 3. Additional information regarding every downstream tasks.
>
> - In the main text, we added a more detailed explanation and cross-references to appendix in Section 3.3 to help readers better understand the technical aspects of model deployment.
>
> The technical clarifications are also added in the general response for your reference. Please let us know if any other clarifications are needed.
>
> ## Q1 - What is dataset-specific pre-training.
> Thank you for your question. We refer to the differences between cross-domain and within-domain pre-training schemes. In both cases, the model is first pre-trained on a pre-training dataset using reconstruction loss, then fine-tuned on a downstream dataset with task-specific loss functions. ‘Dataset-specific pre-training’ refers to scenarios where the pre-training and downstream datasets are identical. This clarification has been included in the revised manuscript.
>
> ## Q2 - Line 123: "trained while maintaining 82% average performance" - average performance on what?
> Thank you for your question. In Table 2, we calculate an average error rate across the three downstream tasks as the final evaluation score. The average performance is reported as the relative difference between these final evaluation scores.
>
> ## Q3 - What is meant by "encountering new channels at test time".
> Thank you for pointing this out. We are referring to scenarios discussed in the literature [1, 2, 3], where new or missing channels appear during the testing stage that were not observed during training. This has been further clarified in the revised manuscript.
>
> [1] McKinzie, B., Shankar, V., Cheng, J. Y., Yang, Y., Shlens, J., & Toshev, A. T. (2023, July). Robustness in multimodal learning under train-test modality mismatch. In International Conference on Machine Learning (pp. 24291-24303). PMLR.
>
> [2] Liu, R., Zippi, E. L., Pouransari, H., Sandino, C., Nie, J., Goh, H., ... & Moin, A. (2023). Frequency-aware masked autoencoders for multimodal pretraining on biosignals. arXiv preprint arXiv:2309.05927.
>
> [3] Ye, J., & Pandarinath, C. (2021). Representation learning for neural population activity with Neural Data Transformers. arXiv preprint arXiv:2108.01210.

---

> > ### Author Response · Authors · 2024-11-28
> > **Response to 3Cgn #2**
> >
> > ## Q4 - Why not test on the forecasting task too?
> > Thank you for your suggestion. While forecasting tasks are currently popular in the community, our study focuses on validating the pre-training methodology, and the authors believe that identifying suitable time series datasets for forecasting from a functional perspective requires a separate, comprehensive investigation, which is beyond the scope of this paper.
> >
> > Regardless of the above, to address your concerns, we performed a preliminary forecasting study to explore how NoTS can be adapted for forecasting tasks. Specifically, we compared the performance of NoTS (next-function prediction) with next-patch prediction transformers, using identical backbone architectures. The evaluation was carried out on three widely used forecasting datasets: ECL, Traffic, and Weather, which have been newly included in our experimental results. We tested two approaches for applying NoTS to forecasting tasks, and the methodology and results are detailed below:
> >
> > ### 1. Using input tokens for forecasting
> >
> > We initialize the model with pre-trained weights and perform full-scale fine-tuning to update the weights of the encoder, transformer, and decoder for the forecasting tasks. We used the channel mixer with a hyperparameter search across mixed channel values of {10, 20, 50}. We feed the tokens generated from the input series into the transformer and decoder without appending additional task tokens, and tune the entire model with the reconstruction loss between the produced output and the forecasting target. In all experiments, the prediction length and the input length were both set as 96. For the forecasting experiments, we deploy the NoTS-lw model with a hidden dimension of 128.
> >
> > | MSE loss    | ECL | Traffic | Weather |
> > | --- | --- | --- | --- |
> > |  Next-pred       | 0.220   | 0.612       | 0.177       |
> > | **NoTS (Ours)**     | **0.175**   | **0.587**       | **0.172**       |
> >
> > ### 2. Appending prompt tokens for forecasting
> >
> > We also evaluated the prompt tuning strategy as previously used in classification tasks on the forecasting task. In this scenario, we append N=10 randomly initialized task-specific tokens to the input sequence and feed the new set of tokens to the transformer, and perform a full-scale fine-tuning strategy similar to what is described above. We report the results as follows:
> >
> > | MSE loss      | ECL | Traffic | Weather |
> > | --- | --- | --- | --- |
> > | Next-pred      | 0.283   | 0.786       | 0.213       |
> > | **NoTS**            | **0.203**   | **0.634**       | **0.175**       |
> >
> > As demonstrated, our proposed pre-training method **NoTS shows promising potential for forecasting**, even though it faces inherent disadvantages compared to next-patch prediction pre-training, which is a more natural pretext task for forecasting. Interestingly, we discovered that appending prompt tokens for forecasting tasks underperforms compared to directly using the input tokens for forecasting, indicating the intrinsic differences between the classification and forecasting tasks. While the current results highlight the potential of NoTS for tackling more complex time series tasks, we view adapting NoTS for forecasting as a separate, meaningful avenue for future study.
> >
> > **Finally, we sincerely appreciate the effort you invested in providing detailed feedback, which has been instrumental in enhancing the quality of our paper. We hope the additional context clarifies our approach and are open to further discussion if needed.**

---

> ### Author Response · Authors · 2024-12-02
>
> We would like to kindly follow up regarding the responses we submitted to your insightful questions and comments. We truly value your feedback and deeply appreciate the time and effort you have dedicated to reviewing our work. If further clarification or additional information is needed from our side, we would be more than happy to assist. Thank you once again for your time and efforts, and we look forward to hearing from you soon.

---

### Official Review · Reviewer_fbrG · 2024-11-02

**Soundness:** 3
**Presentation:** 4
**Contribution:** 3
**Rating:** 3
**Confidence:** 4

**Summary:**

The paper presents a novel approach to time series analysis by treating data as functions of time. It introduces a pre-training framework called NoTS, which uses degradation operators to create augmented data, allowing an autoregressive transformer to learn interrelationships among functions. Experimental results show that NoTS outperforms existing methods across various tasks on synthetic and real-world datasets. Additionally, a lightweight model, NoTS-lw, is introduced, achieving high performance with minimal training parameters. Overall, the work emphasizes the benefits of a functional perspective in enhancing time series modeling.

**Strengths:**

The method shifts the focus from traditional time series modeling to viewing data as functions of time, which enhances the understanding of temporal dynamics. The NoTS framework effectively utilizes degradation operators to create augmented data, improving the learning process for autoregressive transformers.

**Weaknesses:**

While the paper demonstrates the effectiveness of the NoTS framework across various tasks and datasets, the experimental validation is primarily focused on a limited number of synthetic and real-world datasets. This raises questions about the generalizability of the findings. The performance of the proposed method in more diverse and complex scenarios, such as high-dimensional time series or those with significant noise, remains untested. Expanding the range of datasets and tasks could provide a more comprehensive evaluation of the model's robustness and adaptability; The introduction of degradation operators and the functional perspective may complicate the implementation of the NoTS framework. Practitioners may find it challenging to adapt the proposed methods to their specific use cases without a deep understanding of the underlying theory. This complexity could limit the accessibility of the framework for researchers and practitioners who are not well-versed in advanced time series modeling techniques, potentially hindering its adoption in real-world applications; Although the paper discusses the scalability of the NoTS-lw model, it does not provide extensive insights into how the framework performs with significantly larger datasets or more complex models. The scalability of the approach in terms of computational efficiency and memory usage is crucial for practical applications, especially in industries dealing with large-scale time series data. Without thorough analysis and benchmarks on scalability, it is difficult to ascertain whether the proposed methods can be effectively utilized in resource-constrained environments or for large-scale deployments.

**Questions:**

1, How does the NoTS framework compare to traditional time series modeling techniques in terms of interpretability and ease of use for practitioners who may not have a strong background in advanced machine learning methods?

2, What specific strategies could be implemented to enhance the scalability of the NoTS framework for large-scale time series datasets, and how might these strategies impact the performance and computational efficiency of the model?

---

> ### Author Response · Authors · 2024-11-28
> **Response to fbrG**
>
> Thank you for your insightful questions and comments. Below are our responses:
> ## W1 - The experimental validation is primarily focused on a limited number of synthetic and real-world datasets. It is difficult to ascertain whether the proposed methods can be effectively utilized in resource-constrained environments or for large-scale deployments.
>
> Thank you for your feedback. We would like to emphasize the scope of the work as follows:
> - Building on the idea that transformers capture relationships in sequences, our paper proposes and validates the idea that the sequential input to the transformer could be constructed as an alternative functional form, rather than simply segmenting time series into patches.
>
> - In this work, we propose the above idea, theoretically justify it by analyzing transformers’ capacity to approximate functions when constructing different sequences, directly demonstrate its advantages through synthetic experiments for feature approximation, and further validate it on real-world datasets across diverse tasks.
>
> - We have stated that the experimental results on real-world datasets are **an initial attempt** to demonstrate the idea’s feasibility in real life, as noted both in the experimental section (line 518-520) and the limitation section (line 536-538). **In the revised manuscript, we have further tone down the contribution claims in the abstract and introduction**. Our focus is on validating the idea in a theoretically grounded way, rather than conducting large-scale or scaled-up experiments. We believe the current results, combined with theoretical analysis, sufficiently justify the method. While we believe the proposed method provides a viable alternative for building time series foundation models, the goal of this work is not to develop a ChatGPT-like product or similar large-scale application.
>
> ## W2 - The introduction of degradation operators and the functional perspective may complicate the implementation of the NoTS framework.
>
>
> We are happy to improve the presentation for better clarity and have included three new sets of ablation experiments in the revised manuscript. While we acknowledge that the method may appear complex, it becomes straightforward when the underlying mechanism is understood. However, some level of complexity is unavoidable to clearly articulate the theoretical advantages and fully explain the methodology. Additionally, as we aim to develop and demonstrate a novel learning method, certain complexities are inherent to achieving this goal.
>
>
> ## Q3 - How does the NoTS framework compare to traditional time series modeling techniques in terms of interpretability and ease of use for practitioners who may not have a strong background in advanced machine learning methods?
> Thank you for your question. We believe that identifying which degradation operator is most effective for a given dataset can provide valuable insights into the dataset’s underlying characteristics, enhancing interpretability.
>
>
> ## Q4 - What specific strategies could be implemented to enhance the scalability of the NoTS framework for large-scale time series datasets, and how might these strategies impact the performance and computational efficiency of the model?
> Thank you for your question. We believe future studies should focus on the following critical perspectives: (1) Developing more diverse and high-quality synthetic datasets for pre-training; (2) Exploring additional degradation operators, their combinations, and applications to longer sequences. These efforts would require larger-scale and longer training processes, which, based on our initial results, are likely to improve performance. However, these aspects are beyond the scope of the current study and will be addressed in future work.

---

> ### Author Response · Authors · 2024-12-02
>
> We would like to kindly follow up regarding the responses we submitted to your insightful questions and comments. We truly value your feedback and deeply appreciate the time and effort you have dedicated to reviewing our work. If further clarification or additional information is needed from our side, we would be more than happy to assist. Thank you once again for your time and efforts, and we look forward to hearing from you soon.

---

### Official Review · Reviewer_WkQH · 2024-11-02

**Soundness:** 2
**Presentation:** 3
**Contribution:** 2
**Rating:** 5
**Confidence:** 3

**Summary:**

This paper introduces NoTS, a novel pre-training framework that reinterprets time series data through a functional lens. The framework constructs alternative sequences through two types of degradation operators: local smoothing (using averaging kernels with lengths $p_k$) and global smoothing (using sinc functions with frequency cutoff values $0.5p_k$), which create progressively simplified variants of the original signal while preserving functional properties.

The pre-training works by training an autoregressive transformer to recover increasingly detailed versions of the signal. A channel-independent encoder first converts each degraded signal $S_k$ into token groups $R_k$, then the transformer predicts the next resolution level using previous levels, with a decoder reconstructing the predicted signals. The training combines an AR reconstruction loss comparing $S'_{k+1}$ with $S_k$ and a latent consistency term for the original signal.

For real-world deployment, NoTS uses channel adaptors (linear layer, retrainable channel tokens) and task adaptors (prompt tuning, task-specific linear layers) requiring <1% new parameters while maintaining 82% performance. Experiments across 22 datasets spanning classification, anomaly detection, and imputation tasks are evaluated.

**Strengths:**

1. Reformulates time series modeling from a functional perspective and introduces degradation operators to create multiple views.
2. Proposes autoregressive modeling between resolution levels instead of traditional next-time-point prediction.
3. Provides function approximation analysis for transformer limitations on time series.

**Weaknesses:**

1. Limited Scope of Evaluation
- While the paper's focus on generalizable time series modeling, the crucial forecasting task is missing. The next-time-point prediction paradigm has clear connections to dynamic evolution, but the benefits of resolution-based prediction for forecasting tasks remain unexplored. This gap makes it difficult to fully assess the method's practical advantages.
- Performance improvements are moderate except for classification tasks.
- Limited datasets for imputation task. The ETT datasets are homogenous, and the model’s performance should be evaluated across other commonly used datasets as well, like Electricity and Weather.
- Limited ablation studies on key components. It would be better to show the impact of different degradation operators, degradation levels, and frequency cutoff points $p_k$ and diverse real-world datasets as well.
2. Minor parts:
- Technical clarification is needed on loss alignment choices, embedding usage for downstream tasks, and prompt tuning plan.

**Questions:**

1. Which embeddings from the pre-trained model are used for downstream tasks?
2. While following Jia et al. (2022), the adaptation of visual prompt tuning to time series data requires a more detailed explanation. Time series-specific prompt token structure and organization should be elaborated.
3. Please justify the alignment of $S_{k+1}'$ with $S_k$ in the loss function rather than $S_{k+1}$. The alternative of matching with $S_{k+1}$ should be discussed.

---

> ### Author Response · Authors · 2024-11-28
> **Response to WkQH #1**
>
> Thank you for your insightful questions and comments. Below are our responses:
> ## W1 - The crucial forecasting task is missing.
> Thank you for your suggestion. While forecasting tasks are currently popular in the community, our study focuses on validating the pre-training methodology, and the authors believe that identifying suitable time series datasets for forecasting from a functional perspective requires a separate, comprehensive investigation, which is beyond the scope of this paper.
>
> Regardless of the above, to address your concerns, we performed a preliminary forecasting study to explore how NoTS can be adapted for forecasting tasks. Specifically, we compared the performance of NoTS (next-function prediction) with next-patch prediction transformers, using identical backbone architectures. The evaluation was carried out on three widely used forecasting datasets: ECL, Traffic, and Weather, which have been newly included in our experimental results. We tested two approaches for applying NoTS to forecasting tasks, and the methodology and results are detailed below:
> ### 1. Using input tokens for forecasting
>
> We initialize the model with pre-trained weights and perform full-scale fine-tuning to update the weights of the encoder, transformer, and decoder for the forecasting tasks. We used the channel mixer with a hyperparameter search across mixed channel values of {10, 20, 50}. We feed the tokens generated from the input series into the transformer and decoder without appending additional task tokens, and tune the entire model with the reconstruction loss between the produced output and the forecasting target. In all experiments, the prediction length and the input length were both set as 96. For the forecasting experiments, we deploy the NoTS-lw model with a hidden dimension of 128.
>
> | MSE loss    | ECL | Traffic | Weather |
> | --- | --- | --- | --- |
> |  Next-pred       | 0.220   | 0.612       | 0.177       |
> | **NoTS (Ours)**     | **0.175**   | **0.587**       | **0.172**       |
>
> ### 2. Appending prompt tokens for forecasting
>
> We also evaluated the prompt tuning strategy as previously used in classification tasks on the forecasting task. In this scenario, we append N=10 randomly initialized task-specific tokens to the input sequence and feed the new set of tokens to the transformer, and perform a full-scale fine-tuning strategy similar to what is described above. We report the results as follows:
>
> | MSE loss      | ECL | Traffic | Weather |
> | --- | --- | --- | --- |
> | Next-pred      | 0.283   | 0.786       | 0.213       |
> | **NoTS**            | **0.203**   | **0.634**       | **0.175**       |
>
> As demonstrated, our proposed pre-training method **NoTS shows promising potential for forecasting**, even though it faces inherent disadvantages compared to next-patch prediction pre-training, which is a more natural pretext task for forecasting. Interestingly, we discovered that appending prompt tokens for forecasting tasks underperforms compared to directly using the input tokens for forecasting, indicating the intrinsic differences between the classification and forecasting tasks. While the current results highlight the potential of NoTS for tackling more complex time series tasks, we view adapting NoTS for forecasting as a separate, meaningful avenue for future study.
>
> ## W2 - Performance improvements are moderate except for classification tasks. Limited datasets.
> Thank you for your feedback. We would like to emphasize the scope of the work as follows:
> - Building on the idea that transformers capture relationships in sequences, our paper proposes and validates the idea that the sequential input to the transformer could be constructed as an alternative functional form, rather than simply segmenting time series into patches.
>
> - In this work, we propose the above idea, theoretically justify it by analyzing transformers’ capacity to approximate functions when constructing different sequences, directly demonstrate its advantages through synthetic experiments for feature approximation, and further validate it on real-world datasets across diverse tasks.
>
> - We have stated that the experimental results on real-world datasets are **an initial attempt** to demonstrate the idea’s feasibility in real life, as noted both in the experimental section (line 518-520) and the limitation section (line 536-538). **In the revised manuscript, we have further tone down the contribution claims in the abstract and introduction**. Our focus is on validating the idea in a theoretically grounded way, rather than conducting large-scale or scaled-up experiments. We believe the current results, combined with theoretical analysis, sufficiently justify the method. While we believe the proposed method provides a viable alternative for building time series foundation models, the goal of this work is not to develop a ChatGPT-like product or similar large-scale application.

---

> > ### Author Response · Authors · 2024-11-28
> > **Response to WkQH #2**
> >
> > ## W3 - Limited ablation studies on key components.
> > Thank you for your questions, we provide three new sets of additional ablations as below:
> > ### 1. Ablation study on the proportion of global and local degradation operators
> >
> > We conducted an ablation study on the degradation operator by varying the proportions of global and local operators. Using the prompt fine-tuning strategy described in the paper, we pre-trained four new NoTS variants: 25/75, 75/25, and with either global (0/100) or local (100/0) operators removed, benchmarking their performance against the original 50/50 model. We evaluate the model’s performance using the classification tasks performed on the 9 sets of UCR datasets, and the results are as follows:
> >
> > | Global/Local         | 0/100      | 25/75     | 50/50     | 75/25     | 100/0    |
> > | --- | --- | --- | --- | --- | --- |
> > | **Classification AVG**  | **71.60**  | **71.42**  | **71.88**  | **71.23**  | **69.55**  |
> >
> > We draw two conclusions from the results:
> > - **NoTS demonstrates robustness to variations** in the proportion of degradation operators.
> > - The optimal degradation operator appears to be dataset-dependent. For instance, HandOutlines benefits from local degradation, while FordB performs better with global degradation. This suggests **NoTS could serve as a useful tool for exploring dataset properties**.
> >
> > ### 2. Ablation study on the intensity of degradation by adjusting smoothing kernels
> >
> > The models in the paper employed local smoothing kernels with values {2, 4, 8, 16} and global smoothing kernels with frequency cutoff components of {128, 64, 32, 16}. We conducted two new experiments to uniformly decrease or increase the degradation intensity during the pre-training stage, and employ the newly pre-trained models on the same classification task for evaluation. We detail the selected kernels in the table. **For presentation clarity, only the average performance is reported below, while dataset-specific results are provided in the revised manuscript**.
> >
> > | | Decrease intensity | Normal intensity | Increase intensity |
> > | --- | --- | --- | --- |
> > | Global kernels | {256, 128, 64, 32} | {128, 64, 32, 16} | {64, 32, 16, 8} |
> > | Local kernels   | {2, 4, 6, 8}                  | {2, 4, 8, 16}                 | {4, 8, 16, 32}                 |
> > | **Classification AVG** | **68.66**       | **71.88**                         | **72.58**           |
> >
> > We observed that increasing the intensity of degradation operators enhances classification performance, while decreasing it has a negative impact.
> >
> > ### 3. Ablation study on the number of degradation steps
> >
> > In paper, we used a fixed sequence length of 5 (including the raw signal) as the constructed signal functional sequence for the pre-training objective. To examine how the number of degradation steps impacts performance, we followed the same training pipeline and varied the sequence length. The models are evaluated on the same set of classification tasks, and we report the average performance as follows:
> >
> > |     Degradation steps          | 3                  | 5                             | 7                                      |
> > | --- | --- | --- | --- |
> > | Global kernels  | {128, 32}   | {128, 64, 32, 16}   | {128, 64, 32, 16, 8, 4} |
> > | Local kernels   | {2, 8}                 | {2, 4, 8, 16}                      | {2, 4, 8, 16, 32, 64}                     |
> > | **Classification AVG** |  **66.48**           |  **71.88**            | **73.69**            |
> >
> > We found that increasing the constructed sequence length improves classification performance. However, an obvious drawback is that adding degradation steps increases memory costs during pre-training, as the memory usage for each batch scales linearly with the amount of degradation steps, while the memory costs during the inference stage remain unaffected.
> >
> > Aside from the new ablations, we also conducted an ablation using an alternative Gaussian degradation operator in Section 5.3 (line 505-510). **We hope the new results can address the raised questions and clarify how and why the proposed method performs effectively.**

---

> > > ### Author Response · Authors · 2024-11-28
> > > **Response to WkQH #3**
> > >
> > > ## W4 - Technical clarification is needed on loss alignment choices, embedding usage for downstream tasks, and prompt tuning plan.
> > >
> > > Thank you for pointing out the clarification issues of the work. We have improved the clarity of the technical details by incorporating the following texts into our revised manuscript**. This includes:
> > > - In Appendix, we add
> > > 1. A more comprehensive description and explanations of the model deployment technical details;
> > > 2. Mathematical formulation of our prompt-tuning strategy;
> > > 3. Additional information regarding the downstream tasks.
> > > - In the main text, we added a more detailed explanation and cross-references to appendix in Section 3.3 to help readers better understand the technical aspects of model deployment.
> > >
> > > The technical clarifications are also added in the general response for your reference. Please let us know if any other clarifications are needed.
> > >
> > > ## Q5 - Please justify the alignment of Sk+1′ with Sk in the loss function rather than Sk+1. The alternative of matching with Sk+1 should be discussed.
> > > Thank you for your question. We would like to point out that this ablation is conducted in Table 3 on the feature regression task, where we remove the autoregressive modeling as a possible alternative. We believe the next-step alignment increases the difficulty of the pre-training task, encouraging transformers to build connections across different functional alternatives of the same sample, and thus learn better representations.
> > >
> > > **Finally, we sincerely appreciate the effort you invested in providing detailed feedback, which has been instrumental in enhancing the quality of our paper. We hope the additional context clarifies our approach and are open to further discussion if needed.**

---

> ### Author Response · Authors · 2024-12-02
>
> We would like to kindly follow up regarding the responses we submitted to your insightful questions and comments. We truly value your feedback and deeply appreciate the time and effort you have dedicated to reviewing our work. If further clarification or additional information is needed from our side, we would be more than happy to assist. Thank you once again for your time and efforts, and we look forward to hearing from you soon.

---

### Official Review · Reviewer_DXpz · 2024-11-03

**Soundness:** 2
**Presentation:** 2
**Contribution:** 2
**Rating:** 3
**Confidence:** 3

**Summary:**

This paper proposes a pre-training method for time series data called Narratives of Time Series (NoTS). NoTS frames time series modeling by considering time series samples as a sequence of functions rather than a sequence of time periods. NoTS utilizes degradation functions to generate augmented variations of a time series signal. These augmented signals have varying degrees of simplification and together form a sequence that is used to train an autoregressive transformer. The authors provide theoretical justification for framing time series as functional sequences, showing that this approach can approximate a broader class of functions compared to traditional segmentation into periods. The authors validate NoTS on a variety of tasks, including classification, anomaly detection, and imputation, using both synthetic and real-world datasets.

**Strengths:**

- Originality: the notion of "Narrative of Time Series" by predicting the "next function" in a functional decomposition of the time series is interesting and modestly novel. However, it is strongly patterned after the N-BEATS decomposition (see below), and links and connections to this earlier literature should be made.
- Quality: The experimental results are interesting, although the validity of results is called into question (below).
- Clarity: The paper is at times hard to follow, in particular with respect to notational sloppiness and shortcuts taken in detailing the model.

**Weaknesses:**

- Eq. 1 is sloppily written for the case $i=1$
- Section 2.1: there should be more rigor in the notation. The paper mixes discrete and continuous time, and does not specify how discrete time indices map onto functional arguments in continuous time. For instance, the notion of a "calendar" containing this mapping could be introduced.
- Since the functional representation of time series is so important to the paper, Section 2 lacks a survey of the rich literature on the topic, e.g. from the Gaussian processes literature (e.g. https://proceedings.neurips.cc/paper/2007/hash/81e74d678581a3bb7a720b019f4f1a93-Abstract.html ) and classical statistics (e.g. “Functional Data Analysis” by J.O. Ramsay and B.W. Silverman (1997)).
- The proposed decomposition is strongly reminiscent of the N-BEATS approach, which is not acknowledged. The paper should contrast and compare their proposed approach to N-BEATS:
```
@inproceedings{
Oreshkin2020N-BEATS:,
title={N-BEATS: Neural basis expansion analysis for interpretable time series forecasting},
author={Boris N. Oreshkin and Dmitri Carpov and Nicolas Chapados and Yoshua Bengio},
booktitle={International Conference on Learning Representations},
year={2020},
url={https://openreview.net/forum?id=r1ecqn4YwB}
}
```

**Questions:**

- Line 142: "Typically, each dataset has its unique channel-wise relationships" ==> not clear what this sentence means.
- If the discrete-time signal is obtained by sampling at irregular intervals or contains missing data, discuss how convolution operations would work for the degradation function.
- Line 256: "encoder produces groups of tokens from each signal" ==> not clear how the signal is chunked into tokens in the first place. Is it through a PatchTST-like mechanism, or some other approach?
- Lines 264-265: The specifics of this model should be given (perhaps in appendix), or a citation provided. The overall data flow, ranging from tokenization, encoder, decoder is very hand-wavy and hard to follow.
- Lines 282-283: The argument made to explain the addition of the latent consistency term is not clear and should be further explained.
- Section 3.3, test-time adaptors: it is not clear whether these adaptors enable zero-shot handling of new time series, or if adaptor fine-tuning is required.
- Eq. (3) specifies that the absolute error is used as the training loss. This makes the model sensitive to the scale of the time series — if the series are not scale-normalized, training will overweight series with the largest absolute magnitude. Further, section 5.2 does not discuss this issue, which calls into question the validity of experimental results.

---

> ### Author Response · Authors · 2024-11-28
> **Response to DXpz #1**
>
> Thank you for your insightful questions and comments. Below are our responses:
>
> ## W1 - Lack of related literature and similarity with the N-BEATS method.
> First, we would like to thank you for raising those related literature to our attention. The literature is indeed very relevant, and we have added a new section in the revised manuscript to discuss and cite them. We copy our new section as follows for your reference:
> ___
> A line of traditional time series modeling methods focus on learning samples from the functional perspective [1], using statistical approaches [2], or their advanced variations like the Theta method [3] or ARIMA models [4]. These methods have been extended to deep learning through basis expansion approaches, with N-BEATS [5] being a prominent example. N-BEATS uses fully connected layers to perform hierarchical time series decomposition by generating coefficients for predefined or learnable neural bases. N-HiTS [6] builds on N-BEATS by incorporating subsampling layers, enabling multifrequency data sampling and multi-scale interpolation for improved predictions. Our work differs from prior approaches by focusing on the advantages of building sequences with functional awareness through the transformer architecture.
>
> [1] Chapados, N., & Bengio, Y. (2007). Augmented functional time series representation and forecasting with Gaussian processes. Advances in neural information processing systems, 20.
>
> [2] Holt, C. C. (2004). Forecasting seasonals and trends by exponentially weighted moving averages. International journal of forecasting, 20(1), 5-10.
>
> [3] Assimakopoulos, V., & Nikolopoulos, K. (2000). The theta model: a decomposition approach to forecasting. International journal of forecasting, 16(4), 521-530.
>
> [4] Hyndman, R. J., & Khandakar, Y. (2008). Automatic time series forecasting: the forecast package for R. Journal of statistical software, 27, 1-22.
>
> [5] Oreshkin, B. N., Carpov, D., Chapados, N., & Bengio, Y. (2019). N-BEATS: Neural basis expansion analysis for interpretable time series forecasting. arXiv preprint arXiv:1905.10437.
>
> [6] Challu, C., Olivares, K. G., Oreshkin, B. N., Ramirez, F. G., Canseco, M. M., & Dubrawski, A. (2023, June). Nhits: Neural hierarchical interpolation for time series forecasting. In Proceedings of the AAAI conference on artificial intelligence (Vol. 37, No. 6, pp. 6989-6997).
> ___
> However, we would like to point out that the key innovation of our work lies beyond traditional approaches.
> - **The use and analysis of transformers are integral to our work** and represent a non-neglectable contribution, which distinguishes our work from previous works. Building on the idea that transformers capture relationships in sequences, our paper proposes and validates the idea that the sequential input to the transformer could be constructed as an alternative functional form, rather than simply segmenting time series into patches.
> - In this work, we propose the above idea, theoretically justify it by analyzing **transformers’ capacity** to approximate functions when constructing different sequences, directly demonstrate its advantages through synthetic experiments for feature approximation, and further validate it on real-world datasets across diverse tasks.
> Our work highlights the importance of functional analysis approaches in the context of transformers, a perspective that, to the best of our knowledge, has not been explored in previous studies.
>
> ## W2 - Eq. 1 is sloppily written for the case i=1
> Thank you for your feedback. We would like to emphasize that our work considers only cases where $i>1$. Our method is designed specifically for transformers, which require processing token sequences with more than one token.
>
> ## W3 - How discrete time indices map onto functional arguments in continuous time.
> Thank you for your feedback. Our work focuses on mapping functional arguments from continuous time to discrete time through the sampling process and does not address the reverse direction. We would appreciate it if you could suggest relevant works to help us improve the clarity of our paper from this perspective.

---

> > ### Author Response · Authors · 2024-11-28
> > **Response to DXpz #2**
> >
> > ## Q4 - Questions and clarity issues:
> > 1. Line 142: "Typically, each dataset has its unique channel-wise relationships"
> >
> > **Response:** We refer to the fact that pre-training time series datasets may have different channel mappings compared to downstream datasets. To improve clarity, we have added a new paragraph in the revised manuscript and expanded the appendix with sections detailing our model deployment pipeline.
> >
> > 2. If the discrete-time signal is obtained by sampling at irregular intervals or contains missing data, discuss how convolution operations would work for the degradation function.
> >
> > **Response:** Thank you for your feedback. Our work focuses exclusively on regularly sampled time series.
> >
> > 3. Line 256: "encoder produces groups of tokens from each signal" ==> not clear how the signal is chunked into tokens in the first place. Is it through a PatchTST-like mechanism, or some other approach?
> >
> > **Response:** Thank you for your question. Our pre-training strategy NoTS is independent of the specific chosen encoder. For all experiments involving NoTS-lw, we used a 1D channel-independent ResNet encoder/decoder architecture, as specified in main text line 263-266 and Appendix B.2.2. Additionally, in Table 2, we attached our pre-training strategy NoTS on top of PatchTST and iTransformer encoders, demonstrating performance improvements.
> >
> > 4. The overall data flow, ranging from tokenization, encoder, decoder is very hand-wavy and hard to follow.
> >
> > **Response:** Thank you for pointing out the issues. We have added a new section in the Appendix providing all experimental and model deployment details from pre-training to fine-tuning and testing. Please let us know if there is any other clarification needed.
> >
> > 5. Section 3.3, test-time adaptors: it is not clear whether these adaptors enable zero-shot handling of new time series, or if adaptor fine-tuning is required.
> >
> > **Response:** Thank you for your question. The adaptors are used in our context-aware adaptation scheme, where the adaptors are updated during the fine-tuning stage in a parameter efficient manner. We have provided additional details in Appendix C.1 and also the general response, please let us know if additional clarifications are needed.
> >
> > 6. Lines 282-283: The argument made to explain the addition of the latent consistency term is not clear and should be further explained.
> >
> > **Response:** The autoregressive framework predicts complex functions through their simplified variants, training the model with reconstruction loss across different degradations. For example, given original data $\mathbf{X}$, a simplified variant $\mathbf{X}_1$ is generated, and $\mathbf{X}_1$ is used to predict $\mathbf{X}$. Here, the encoder does not need to extract the latent representation of $\mathbf{X}$ since the transformer will perform the prediction of the next step. To address this, latent consistency loss is applied to ensure the encoder captures the raw representation of $\mathbf{X}$. For further clarity, please refer to the top row of Figure 2.
> >
> > 7. The scale of the time series.
> >
> > **Response:** Thank you for pointing this out. We followed the data normalization pipeline from previous works, ensuring all data is normalized before being passed into the encoder. A detailed description of our data preprocessing pipeline has been added to the Appendix.
> >
> > We sincerely appreciate the effort you put into providing detailed feedback, which has been invaluable in improving the clarity of our paper. However, we believe there are significant areas where our intent may not have been fully understood, particularly regarding the use of transformers in this problem and their role in connecting different functional representations. These aspects represent the core contributions of our work. We hope the additional context provided clarifies our approach and are happy to discuss further if needed. We kindly ask you to reconsider your score after reviewing our responses.

---

> ### Author Response · Authors · 2024-12-02
>
> We would like to kindly follow up regarding the responses we submitted to your insightful questions and comments. We truly value your feedback and deeply appreciate the time and effort you have dedicated to reviewing our work. If further clarification or additional information is needed from our side, we would be more than happy to assist.
> Thank you once again for your time and efforts, and we look forward to hearing from you soon.

---

### Official Review · Reviewer_tTyA · 2024-11-03

**Soundness:** 3
**Presentation:** 3
**Contribution:** 3
**Rating:** 8
**Confidence:** 3

**Summary:**

This paper presents Narratives of Time Series (NoTS), a transformer-based framework that reinterprets time series data as temporal functions, capturing both local and global properties of time series through a novel pre-training objective. The authors introduce three main innovations:

Progressive Functional Degradation: NoTS generates simplified versions of time series by applying degradation operators that progressively smooth the signal. Each degradation level creates a coarser function, enabling the model to learn time series as a sequence of simplified functions rather than individual time points, preserving the functional structure over time.

Autoregressive Transformer with Next-Function Prediction: Inspired by next-word prediction in NLP, NoTS trains the transformer to reconstruct the original time series from its degraded variants. By predicting the next function in a sequence, each step adds more information, allowing the model to capture nuanced temporal dependencies across multiple scales.

Adaptation via Channel and Task Adaptors: NoTS includes adaptable modules to handle different time series tasks and varying channel structures, which improves model generalization and reusability across datasets. These adaptors support context-aware adaptation and help the model perform well with minimal parameter updates, making it versatile across tasks.

Experiments and Analysis:
The authors begin by presenting a theoretical analysis that shows standard transformers struggle to generalize when applied to time series tasks that involve complex operations, such as differentiation, due to their reliance on discretized data. Specifically, the authors prove that transformers encounter high approximation errors when attempting to model certain mathematical operations (e.g., differentiation) on time series. Authors also prove that their proposed method viewing time series data as continuous functions in "function space" rather than as discrete can enable the transformer to better capture the underlying properties of the time series.

NoTS is evaluated on both synthetic and real-world time series datasets, covering tasks such as classification, imputation, and anomaly detection. NoTS demonstrates strong performance across diverse datasets, outperforming established baselines such as masked autoencoders and next-period prediction models. Ablation studies further validate the importance of each component, and scalability experiments indicate that NoTS’s performance improves predictably with model size, in line with power-law trends seen in other fields like NLP.

**Strengths:**

- The borrowed functional narrative approach is innovative for time series modality, addressing key limitations in time series modeling due to discretization. By leveraging function-based degradations, NoTS effectively captures temporal relationships, making it a strong framework for autoregressive analysis.

- Clear theoretical grounding in approximation theory supports the functional degradation and autoregressive prediction framework. Visualizations (e.g., degradation sequences) help convey complex concepts effectively.

- Experimental results show NoTS’s robustness across diverse tasks and datasets, with context-aware adaptors allowing minimal retraining. These adaptors facilitate handling different channel mappings and task-specific nuances, enhancing practical application.

**Weaknesses:**

- The paper lacks an analysis of how sensitive NoTS is to the choice of the degradataion operators or guidance on selecting appropriate ones. This dependence could impact performance if the degradations are not well-suited to certain datasets or applications. The ablation study could be improved by including more ablation tests on this aspect.

- While NoTS is evaluated on classification, anomaly detection, and imputation, it lacks experiments on long-horizon forecasting tasks, which are crucial for assessing a model’s ability to capture long-range dependencies and temporal features (as highlihted in the intro). This addition would also reinforce the paper’s motivation, presented in the introduction, which critiques patch-based transformer architectures focused on forecasting tasks (e.g., Moirai and Lagllama).

**Questions:**

- Could the authors provide an analysis on how sensitive NoTS is to the choice of degradation operators? Specifically, how does the performance vary when different types of degradation operators are used, or when the balance between local and global smoothing is adjusted?
- Is there any guidance the authors could offer on selecting suitable degradation operators for different datasets or applications? A more detailed ablation on this aspect would clarify the robustness of NoTS across varying conditions.
- Is there any limitations sourced from the pretraining objective that stops authors testing on the forecasting task? If not do the authors plan to evaluate NoTS on long-horizon forecasting tasks? Given that capturing long-range dependencies is highlighted as a core strength in the introduction, an assessment on forecasting (especially long-range) would provide valuable insights.
- Could the authors provide an analysis of how NoTS’s performance contribution varies with increasing time series lengths? Understanding how NoTS scales with longer time series would be valuable for assessing its capability to capture long-range dependencies and its potential applications to extended temporal data.

---

> ### Author Response · Authors · 2024-11-28
> **Response to tTyA #1**
>
> Thank you for your insightful questions and comments. Below are our responses:
> ## W1 - New ablation studies regarding “the choice of the degradation operators” and “when the balance between local and global smoothing is adjusted”.
> We provide three new sets of ablation experiments following your suggestions. The results are also detailed in the general response, but we also copied them here for your convenience:
> ### 1. Ablation study on the proportion of global and local degradation operators
>
> We conducted an ablation study on the degradation operator by varying the proportions of global and local operators. Using the prompt fine-tuning strategy described in the paper, we pre-trained four new NoTS variants: 25/75, 75/25, and with either global (0/100) or local (100/0) operators removed, benchmarking their performance against the original 50/50 model. We evaluate the model’s performance using the classification tasks performed on the 9 sets of UCR datasets, and the results are as follows:
>
> | Global/Local         | 0/100      | 25/75     | 50/50     | 75/25     | 100/0    |
> | --- | --- | --- | --- | --- | --- |
> | **Classification AVG**  | **71.60**  | **71.42**  | **71.88**  | **71.23**  | **69.55**  |
>
> We draw two conclusions from the results:
> - **NoTS demonstrates robustness to variations** in the proportion of degradation operators.
> - The optimal degradation operator appears to be dataset-dependent. For instance, HandOutlines benefits from local degradation, while FordB performs better with global degradation. This suggests **NoTS could serve as a useful tool for exploring dataset properties**.
>
> ### 2. Ablation study on the intensity of degradation by adjusting smoothing kernels
>
> The models in the paper employed local smoothing kernels with values {2, 4, 8, 16} and global smoothing kernels with frequency cutoff components of {128, 64, 32, 16}. We conducted two new experiments to uniformly decrease or increase the degradation intensity during the pre-training stage, and employ the newly pre-trained models on the same classification task for evaluation. We detail the selected kernels in the table. **For presentation clarity, only the average performance is reported below, while dataset-specific results are provided in the revised manuscript**.
>
> | | Decrease intensity | Normal intensity | Increase intensity |
> | --- | --- | --- | --- |
> | Global kernels | {256, 128, 64, 32} | {128, 64, 32, 16} | {64, 32, 16, 8} |
> | Local kernels   | {2, 4, 6, 8}                  | {2, 4, 8, 16}                 | {4, 8, 16, 32}                 |
> | **Classification AVG** | **68.66**       | **71.88**                         | **72.58**           |
>
> We observed that increasing the intensity of degradation operators enhances classification performance, while decreasing it has a negative impact.
>
> ### 3. Ablation study on the number of degradation steps
>
> In paper, we used a fixed sequence length of 5 (including the raw signal) as the constructed signal functional sequence for the pre-training objective. To examine how the number of degradation steps impacts performance, we followed the same training pipeline and varied the sequence length. The models are evaluated on the same set of classification tasks, and we report the average performance as follows:
>
> |     Degradation steps          | 3                  | 5                             | 7                                      |
> | --- | --- | --- | --- |
> | Global kernels  | {128, 32}   | {128, 64, 32, 16}   | {128, 64, 32, 16, 8, 4} |
> | Local kernels   | {2, 8}                 | {2, 4, 8, 16}                      | {2, 4, 8, 16, 32, 64}                     |
> | **Classification AVG** |  **66.48**           |  **71.88**            | **73.69**            |
>
> We found that increasing the constructed sequence length improves classification performance. However, an obvious drawback is that adding degradation steps increases memory costs during pre-training, as the memory usage for each batch scales linearly with the amount of degradation steps, while the memory costs during the inference stage remain unaffected.

---

> > ### Author Response · Authors · 2024-11-28
> > **Response to tTyA #2**
> >
> > ## W2 - Testing on the forecasting tasks.
> > Thank you for your suggestion. While forecasting tasks are currently popular in the community, our study focuses on validating the pre-training methodology, and the authors believe that identifying suitable time series datasets for forecasting from a functional perspective requires a separate, comprehensive investigation, which is beyond the scope of this paper.
> >
> > Regardless of the above, to address your concerns, we performed a preliminary forecasting study to explore how NoTS can be adapted for forecasting tasks. Specifically, we compared the performance of NoTS (next-function prediction) with next-patch prediction transformers, using identical backbone architectures. The evaluation was carried out on three widely used forecasting datasets: ECL, Traffic, and Weather, which have been newly included in our experimental results. We tested two approaches for applying NoTS to forecasting tasks, and the methodology and results are detailed below:
> >
> > ### 1. Using input tokens for forecasting
> >
> > We initialize the model with pre-trained weights and perform full-scale fine-tuning to update the weights of the encoder, transformer, and decoder for the forecasting tasks. We used the channel mixer with a hyperparameter search across mixed channel values of {10, 20, 50}. We feed the tokens generated from the input series into the transformer and decoder without appending additional task tokens, and tune the entire model with the reconstruction loss between the produced output and the forecasting target. In all experiments, the prediction length and the input length were both set as 96. For the forecasting experiments, we deploy the NoTS-lw model with a hidden dimension of 128.
> >
> > | MSE loss    | ECL | Traffic | Weather |
> > | --- | --- | --- | --- |
> > |  Next-pred       | 0.220   | 0.612       | 0.177       |
> > | **NoTS (Ours)**     | **0.175**   | **0.587**       | **0.172**       |
> >
> > ### 2. Appending prompt tokens for forecasting
> >
> > We also evaluated the prompt tuning strategy as previously used in classification tasks on the forecasting task. In this scenario, we append N=10 randomly initialized task-specific tokens to the input sequence and feed the new set of tokens to the transformer, and perform a full-scale fine-tuning strategy similar to what is described above. We report the results as follows:
> >
> > | MSE loss      | ECL | Traffic | Weather |
> > | --- | --- | --- | --- |
> > | Next-pred      | 0.283   | 0.786       | 0.213       |
> > | **NoTS**            | **0.203**   | **0.634**       | **0.175**       |
> >
> > As demonstrated, our proposed pre-training method **NoTS shows promising potential for forecasting**, even though it faces inherent disadvantages compared to next-patch prediction pre-training, which is a more natural pretext task for forecasting. Interestingly, we discovered that appending prompt tokens for forecasting tasks underperforms compared to directly using the input tokens for forecasting, indicating the intrinsic differences between the classification and forecasting tasks. While the current results highlight the potential of NoTS for tackling more complex time series tasks, we view adapting NoTS for forecasting as a separate, meaningful avenue for future study.
> >
> > ## Q3 - How NoTS’s performance contribution varies with increasing time series lengths?
> > Thank you for your question. We analyzed the performance gain between our next-function prediction method and traditional next-patch prediction method on classification tasks across datasets of varying lengths. The datasets were categorized into three groups: <500 has 4 datasets (PEMS-SF, Heartbeat, ECG5000, and Wafer), 500-1000 has 5 datasets (FordA, FordB, ScreenType, UWaveGestureLibraryAll, SelfRegulationSCP1), and >1000 has 5 datasets (StarLightCurves, HandOutlines, EthanolLevel, EthanolConcentration, SelfRegulationSCP2).
> > | Series length | <500 | 500-1000 | >1000 |
> > | --- | --- | --- | --- |
> > | Relative avg improvement            | 4.82\%  | 0.15\% | 13.04\% |
> >
> > Note that prediction performance can vary significantly across datasets. A more comprehensive study using synthetic datasets could provide deeper insights in the future.
> >
> > Again, we would like to thank you for your valuable feedback. Let us know if there are any other concerns or questions, we are happy to answer!

---

> ### Author Response · Authors · 2024-12-02
>
> We would like to kindly follow up regarding the responses we submitted to your insightful questions and comments. We truly value your feedback and deeply appreciate the time and effort you have dedicated to reviewing our work. If further clarification or additional information is needed from our side, we would be more than happy to assist. Thank you once again for your time and efforts, and we look forward to hearing from you soon.

---

### Author Response · Authors · 2024-11-28
**General response #1 - Scope of the work**

We sincerely thank the reviewers for their valuable feedback.

We are especially encouraged by reviewers’ recognition of the innovation and theoretical results provided by the work. We are particularly grateful to reviewer tTyA for highlighting our contribution as **innovative, addressing key limitations**, reviewer 3Cgn for describing our work **novel, very interesting, bring a very different perspective**. Even reviewers with critical feedback summarized our work as novel (WkQH and fbrG) or interesting (DXpz). Additionally, we really appreciate the positive remarks from reviewer tTyA and 3Cgn on the theoretical rigor of our approach, describing our work as **theoretically grounded** and **well-theoretically justified**.

## Scope of the work

Before presenting additional experimental results, we would like to address some major misunderstandings about the scope and central ideas of our work as follows:

- **The use and analysis of transformers are integral to our work** and represent a non-neglectable contribution, which distinguish our work from previous works. Building on the idea that transformers capture relationships in sequences, our paper proposes and validates the idea that the sequential input to the transformer could be constructed as an alternative functional form, rather than simply segmenting time series into patches.

- In this work, we propose the above idea, theoretically justify it by analyzing transformers’ capacity to approximate functions when constructing different sequences, directly demonstrate its advantages through synthetic experiments for feature approximation, and further validate it on real-world datasets across diverse tasks.

- We have stated that the experimental results on real-world datasets are **an initial attempt** to demonstrate the idea’s feasibility in real life, as noted both in the experimental section (line 518-520) and the limitation section (line 536-538). **In the revised manuscript, we have further tone down the contribution claims in the abstract and introduction**. Our focus is on validating the idea in a theoretically grounded way, rather than conducting large-scale or scaled-up experiments. We believe the current results, combined with theoretical analysis, sufficiently justify the method. While we believe the proposed method provides a viable alternative for building time series foundation models, the goal of this work is not to develop a ChatGPT-like product or similar large-scale application.

- Our work focuses on time series analysis, with the autoregressive objective operating in the functional domain as the training objective. **Forecasting tasks are not the primary focus of this study**, nor do we place significant emphasis on them. While forecasting tasks are currently popular in the community, we believe that identifying suitable time series datasets for forecasting requires a separate, comprehensive investigation, which is beyond the scope of this paper.

We have revised the introduction and abstract sections in the manuscript to help clarify the above misunderstandings for future readers.

In the following responses, we address the key criticisms by providing:
1. Three sets of additional ablation results.
2. Initial forecasting results comparing our next-function prediction with traditional next-patch prediction pre-training methods.
3. Clarifications and detailed descriptions of our experimental setup.

---

> ### Author Response · Authors · 2024-11-28
> **General response #2 - Additional ablation results**
>
> A key criticism of the work is the lack of ablation on the degradation operators for constructing functional sequences. This concern was raised by reviewer tTyA “choice of the degradation operators”, and WkQH “Limited ablation studies on key components”. **We acknowledge the importance of this issue and have spent a decent efforts conducting additional experiments to address this criticism, presenting three new sets of ablation studies**:
>
> ### 1. Ablation study on the proportion of global and local degradation operators
>
> We conducted an ablation study on the degradation operator by varying the proportions of global and local operators. Using the prompt fine-tuning strategy described in the paper, we pre-trained four new NoTS variants: 25/75, 75/25, and with either global (0/100) or local (100/0) operators removed, benchmarking their performance against the original 50/50 model. We evaluate the model’s performance using the classification tasks performed on the 9 sets of UCR datasets, and the results are as follows:
>
> | Global/Local         | 0/100      | 25/75     | 50/50     | 75/25     | 100/0    |
> | --- | --- | --- | --- | --- | --- |
> | HandOutlines   | 88.11  | 72.16  | 71.62  | 71.62  | 64.32  |
> | EthanolLevel   | 29.00    | 28.60   | 28.60   | 28.60   | 26.60  |
> | StarLightCurves| 88.59  | 89.84  | 91.66  | 91.66  | 89.84  |
> | UWave-GL-All   | 65.32  | 64.15  | 67.39  | 57.87  | 56.28  |
> | FordA          | 80.22  | 81.89  | 81.52  | 87.20   | 80.38  |
> | FordB          | 64.57  | 68.52  | 68.27  | 70.00     | 74.07  |
> | Wafer          | 94.11  | 99.61  | 98.78  | 98.83  | 96.38  |
> | ECG5000        | 91.31  | 92.96  | 91.31  | 91.31  | 94.09  |
> | ScreenType     | 43.19  | 45.07  | 47.73  | 44.00     | 44.00     |
> | **Classification AVG**  | **71.60**  | **71.42**  | **71.88**  | **71.23**  | **69.55**  |
>
> We draw two conclusions from the results:
> - **NoTS demonstrates robustness to variations** in the proportion of degradation operators.
> - The optimal degradation operator appears to be dataset-dependent. For instance, HandOutlines benefits from local degradation, while FordB performs better with global degradation. This suggests **NoTS could serve as a useful tool for exploring dataset properties**.
>
> ### 2. Ablation study on the intensity of degradation by adjusting smoothing kernels
>
> The models in the paper employed local smoothing kernels with values {2, 4, 8, 16} and global smoothing kernels with frequency cutoff components of {128, 64, 32, 16}. We conducted two new experiments to uniformly decrease or increase the degradation intensity during the pre-training stage, and employ the newly pre-trained models on the same classification task for evaluation. We detail the selected kernels in the table. **For presentation clarity, only the average performance is reported below, while dataset-specific results are provided in the revised manuscript**.
>
> | | Decrease intensity | Normal intensity | Increase intensity |
> | --- | --- | --- | --- |
> | Global kernels | {256, 128, 64, 32} | {128, 64, 32, 16} | {64, 32, 16, 8} |
> | Local kernels   | {2, 4, 6, 8}                  | {2, 4, 8, 16}                 | {4, 8, 16, 32}                 |
> | **Classification AVG** | **68.66**       | **71.88**                         | **72.58**           |
>
> We observed that increasing the intensity of degradation operators enhances classification performance, while decreasing it has a negative impact.
>
> ### 3. Ablation study on the number of degradation steps
>
> In paper, we used a fixed sequence length of 5 (including the raw signal) as the constructed signal functional sequence for the pre-training objective. To examine how the number of degradation steps impacts performance, we followed the same training pipeline and varied the sequence length. The models are evaluated on the same set of classification tasks, and we report the average performance as follows:
>
> |     Degradation steps          | 3                  | 5                             | 7                                      |
> | --- | --- | --- | --- |
> | Global kernels  | {128, 32}   | {128, 64, 32, 16}   | {128, 64, 32, 16, 8, 4} |
> | Local kernels   | {2, 8}                 | {2, 4, 8, 16}                      | {2, 4, 8, 16, 32, 64}                     |
> | **Classification AVG** |  **66.48**           |  **71.88**            | **73.69**            |
>
> We found that increasing the constructed sequence length improves classification performance. However, an obvious drawback is that adding degradation steps increases memory costs during pre-training, as the memory usage for each batch scales linearly with the amount of degradation steps, while the memory costs during the inference stage remain unaffected.
>
> Aside from the new ablations, we also conducted an ablation using an alternative Gaussian degradation operator in Section 5.3 (line 505-510). **We hope the new results can address the raised questions and clarify how and why the proposed method performs effectively.**

---

> > ### Author Response · Authors · 2024-11-28
> > **General response #3 - Preliminary exploration on forecasting tasks**
> >
> > Although forecasting is not the focus of this work, several reviewers (tTyA, WkQH, 3Cgn) expressed interest in how NoTS would perform on forecasting tasks. **To address this curiosity, we conducted preliminary experiments to compare the performance of NoTS (next-function prediction) against the next-patch prediction transformers using identical backbone architectures.** We evaluated the models’ performance by conducting new experiments on three commonly used forecasting datasets ECL, Traffic, and Weather, which are newly introduced to our experimental results. We explored two approaches for applying NoTS on the forecasting tasks, and report the methodology and results as follows:
> >
> > ### 1. Using input tokens for forecasting
> >
> > We initialize the model with pre-trained weights and perform full-scale fine-tuning to update the weights of the encoder, transformer, and decoder for the forecasting tasks. We used the channel mixer with a hyperparameter search across mixed channel values of {10, 20, 50}. We feed the tokens generated from the input series into the transformer and decoder without appending additional task tokens, and tune the entire model with the reconstruction loss between the produced output and the forecasting target. In all experiments, the prediction length and the input length were both set as 96. For the forecasting experiments, we deploy the NoTS-lw model with a hidden dimension of 128.
> >
> > | MSE loss    | ECL | Traffic | Weather |
> > |----------------------|---------|-------------|-------------|
> > |  Next-pred       | 0.220   | 0.612       | 0.177       |
> > | **NoTS (Ours)**     | **0.175**   | **0.587**       | **0.172**       |
> >
> > ### 2. Appending prompt tokens for forecasting
> >
> > We also evaluated the prompt tuning strategy as previously used in classification tasks on the forecasting task. In this scenario, we append N=10 randomly initialized task-specific tokens to the input sequence and feed the new set of tokens to the transformer, and perform a full-scale fine-tuning strategy similar to what is described above. We report the results as follows:
> >
> > | MSE loss      | ECL | Traffic | Weather |
> > |----------------------|---------|-------------|-------------|
> > | Next-pred      | 0.283   | 0.786       | 0.213       |
> > | **NoTS**            | **0.203**   | **0.634**       | **0.175**       |
> >
> > As demonstrated, our proposed pre-training method **NoTS shows promising potential for forecasting**, even though it faces inherent disadvantages compared to next-patch prediction pre-training, which is a more natural pretext task for forecasting. Interestingly, we discovered that appending prompt tokens for forecasting tasks underperforms compared to directly using the input tokens for forecasting, indicating the intrinsic differences between the classification and forecasting tasks. While the current results highlight the potential of NoTS for tackling more complex time series tasks, we view adapting NoTS for forecasting as a separate, meaningful avenue for future study.

---

> > > ### Author Response · Authors · 2024-11-28
> > > **General response #4 - Clarity issues and technical details**
> > >
> > > Another key criticism of the presented work is the lack of experimental details included in the main text. Since our study follows the data processing pipeline and evaluation protocol of previous works, we moved a lot of details in the appendix due to space constraints. This caused confusion for readers not familiar with relevant literature, as pointed out by reviewer 3Cgn “paper should be self-contained”, WkQH “Technical clarification is needed”, DXpz “shortcuts taken in detailing the model”.
> > >
> > > **Thanks to the reviewers’ feedback, we have improved the clarity of the technical details by incorporating the following texts into our revised manuscript**. This includes:
> > > - In Appendix, we add
> > > 1. A more comprehensive description and explanations of the model deployment technical details;
> > > 2. Mathematical formulation of our prompt-tuning strategy;
> > > 3. Additional information regarding the downstream tasks.
> > > - In the main text, we added a more detailed explanation and cross-references to appendix in Section 3.3 to help readers better understand the technical aspects of model deployment.
> > >
> > > The technical clarifications are also added below for reference:
> > >
> > > ## 1. Overall Pre-training, Fine-tuning, and Testing Pipeline
> > >
> > > ### Overall Model Deployment Flow
> > >
> > > The proposed NoTS method, as a pre-training strategy, considers a pre-training dataset $D_{\text{PT}}$ and a downstream dataset $D_{\text{FT}}$. These datasets have sample dimensions $C_{\text{PT}} \times T_{\text{PT}}$ and $C_{\text{FT}} \times T_{\text{FT}}$, respectively, allowing for differing channel and temporal dimensions between this two phase.
> > >
> > > The model is first pre-trained by performing the autoregressive reconstruction task, guided by the training objective as detailed in Equation 3. It is later fine-tuned on the training split of the downstream dataset, and is finally evaluated on the testing split of the downstream dataset. Depending on the differences between pre-training and downstream datasets, there are two schemes:
> > >
> > > | | Training | Tuning | Testing |
> > > | --- | --- | --- | --- |
> > > | Cross-domain | Synthetic datasets | Real-world datasets (train split) | Real-world datasets (test split) |
> > > | In-domain | Real-world datasets (train split) | Real-world datasets (train split) | Real-world datasets (test split) |
> > >
> > > During pre-training, the input time series are sequentially processed by the encoding layers, the autoregressive transformer, and the decoding layers for the reconstruction task. In fine-tuning, randomly initialized channel adapters and prompt task adapters are added, either trained independently or jointly with the pre-trained encoder, decoder, and transformer during the prompt tuning and full-scale fine-tuning stages, respectively. Finally, the tuned model, with all parameters frozen, is used for testing. The process is detailed below:
> > >
> > > | | Training | Tuning | Testing |
> > > | --- | --- | --- | --- |
> > > | Prompt tuning | Encoder/Transformer/Decoder | Adaptors | NA |
> > > | Full-scale tuning | Encoder/Transformer/Decoder | Encoder/Transformer/Decoder/Adaptors | NA |
> > >
> > > ### Domain shifts across pre-training and downstream datasets
> > >
> > > Domain shift issues between pre-training and downstream datasets are discussed in many recent works on pre-training methods for time series data [1, 2]. These challenges primarily stem from differences between $D_{\text{PT}}$ and $D_{\text{FT}}$, leading to the following two schemes:
> > >
> > > **Cross-domain pre-training and fine-tuning** All NoTS-lw models are trained and evaluated using the cross-domain scheme (first 8 rows of Table 2). Specifically, the models are pre-trained on the synthetic datasets detailed in Section 5.1 to learn and extract the universal dynamic features, then tuned on real-world datasets to demonstrate the generalization capability of the pre-training approach.
> > >
> > >
> > > **Within-domain pre-training and fine-tuning** All NoTS models are trained and evaluated using the within-domain scheme (Table 1 and last 4 rows of Table 2). This setting assesses the model's ability to learn dataset-specific patterns by pre-training and fine-tuning on the same train split of a dataset. The final evaluation of the downstream task in fine-tuning is evaluated on the test split of the corresponding dataset.
> > >
> > > [1] Dong, J., Wu, H., Zhang, H., Zhang, L., Wang, J., & Long, M. (2024). Simmtm: A simple pre-training framework for masked time-series modeling. Advances in Neural Information Processing Systems, 36.
> > >
> > > [2] Liu, R., Zippi, E. L., Pouransari, H., Sandino, C., Nie, J., Goh, H., ... & Moin, A. (2023). Frequency-aware masked autoencoders for multimodal pretraining on biosignals. arXiv preprint arXiv:2309.05927.

---

> > > > ### Author Response · Authors · 2024-11-28
> > > > **General response #5 - Clarity issues and technical details**
> > > >
> > > > ### Prompt tuning and full-scale tuning
> > > >
> > > > We design the prompt tuning method to adapt pre-trained knowledge to new tasks while introducing minimal additional parameters. In this tuning paradigm, the pre-trained transformer (along with the encoder and decoder) remains frozen. Only the adaptor parameters (channel and task adaptors) and, for discriminative tasks, a final linear classification head are trained.
> > > >
> > > > In contrast, the full-scale fine-tuning method unfreezes the pre-trained weights, allowing the entire model, including the encoder, transformer layers, the decoder, the adaptors, and the final linear classification head, to be fine-tuned. This approach enables comprehensive adaptation to new data and tasks but comes with higher computational demands.
> > > >
> > > > ### Testing schemes
> > > >
> > > > The model is tested on the testing split of the downstream dataset. The specific testing scheme varies across different downstream tasks, with major differences across generative (anomaly detection and imputation tasks) and discriminative tasks (classification and regression tasks). More details are included in the downstream tasks section.
> > > >
> > > > ## 2. Mathematical formulation of our prompt-tuning strategy
> > > >
> > > > In the revised manuscript, we specify the methods for our prompt tuning pipeline as follows:
> > > >
> > > > ### Task Adaptors Inspired by Visual Prompt Tuning (VPT)
> > > >
> > > > Consider the input sequence to transformer layer $\mathbf{X} = [\mathbf{x}_1, \mathbf{x}_2, \ldots, \mathbf{x}_N] \in \mathbb{R}^{N \times D}$, where $N$ is the sequence length and $D$ is the transformer's hidden dimension. Our prompt tuning pipeline deploys the deep visual prompt tuning strategy as follows:
> > > >
> > > > We randomly initialize a set of $k$ learnable prompt tokens $\{p_1, \ldots, p_k\}$, where each $p_i \in \mathbb{R}^D$. For each transformer layer $d$, suppose we have $H$ attention heads, and we initialize three separate linear projection layers for the prompt tokens:
> > > >
> > > > $$
> > > > L_{q_p}^{(d)}, \quad L_{k_p}^{(d)}, \quad L_{v_p}^{(d)} \in \mathbb{R}^{D \times (H \times D)},
> > > > $$
> > > >
> > > > each without bias. These layers project the prompt tokens into query, key, and value representations of shape $\mathbb{R}^{H \times D}$ as follows:
> > > >
> > > > $$
> > > > p_{q_i}^{(d)} = p_i L_{q_p}^{(d)}, \quad p_{k_i}^{(d)} = p_i L_{k_p}^{(d)}, \quad p_{v_i}^{(d)} = p_i L_{v_p}^{(d)}, \quad \text{for } i = 1, 2, \ldots, k.
> > > > $$
> > > >
> > > > While the input tokens $\mathbf{x}_j \in \mathbb{R}^D$, where $j = 1, 2, \ldots, N$ have separate linear projection layers that are inherited from the pre-trained transformer attention blocks, giving $\mathbf{x}\_{q\_j}^{(d)}$, $\mathbf{x}\_{k\_j}^{(d)}$, and $\mathbf{x}\_{v\_j}^{(d)}$ for each $j$ in each transformer layer $d$. The projected prompt tokens and input tokens are then concatenated separately to form the augmented query, key, and value sequences:
> > > >
> > > > $$
> > > > \mathbf{Q}\_{\text{aug}}^{(d)} = [p\_{q_1}^{(d)}, p\_{q_2}^{(d)}, \ldots, p\_{q_k}^{(d)}, \mathbf{x}\_{q_1}^{(d)}, \mathbf{x}\_{q_2}^{(d)}, \ldots, \mathbf{x}\_{q_N}^{(d)}],
> > > > $$
> > > >
> > > > $$
> > > > \mathbf{K}\_{\text{aug}}^{(d)} = [p\_{k_1}^{(d)}, p\_{k_2}^{(d)}, \ldots, p\_{k_k}^{(d)}, \mathbf{x}\_{k_1}^{(d)}, \mathbf{x}\_{k_2}^{(d)}, \ldots, \mathbf{x}\_{k_N}^{(d)}],
> > > > $$
> > > >
> > > > $$
> > > > \mathbf{V}\_{\text{aug}}^{(d)} = [p\_{v_1}^{(d)}, p\_{v_2}^{(d)}, \ldots, p\_{v_k}^{(d)}, \mathbf{x}\_{v_1}^{(d)}, \mathbf{x}\_{v_2}^{(d)}, \ldots, \mathbf{x}\_{v_N}^{(d)}].
> > > > $$
> > > >
> > > > These augmented query, key, and value sequences are then processed by the multi-head attention mechanism with $H$ heads in transformer layer $d$, allowing the model to integrate information from both the prompt tokens and the input time series tokens.
> > > >
> > > >
> > > > ### Channel Adaptors
> > > >
> > > > The channel embedding layer is applied before feeding data into the encoding layers, transforming the channel dimension from $C$ to $C'$ as follows:
> > > > $
> > > > L(\mathbf{S}) = W\_{\text{embed}} \mathbf{X} + \mathbf{b}\_{\text{embed}},
> > > > $
> > > > where $W\_{\text{embed}} \in \mathbb{R}^{C' \times C}$ and $\mathbf{b}\_{\text{embed}} \in \mathbb{R}^{C'}$ are learnable parameters.
> > > >
> > > > Additionally, the learnable channel tokens $\mathbf{T} \in \mathbb{R}^{C \times D}$ are used as additive embeddings before the input sequence enters the transformer layers. These tokens recalibrate channel-specific representations, ensuring consistent adjustments across all tokens. This method enhances robustness and flexibility across diverse domains and channel configurations without requiring significant architectural changes.

---

> > > > > ### Author Response · Authors · 2024-11-28
> > > > > **General response #6 - Clarity issues and technical details**
> > > > >
> > > > > ## 3. Additional information for downstream tasks
> > > > >
> > > > > We follow the multitask evaluation pipeline in [3] and provide the necessary details as follows.
> > > > >
> > > > > **Classification** We consider the sequence-level classification task, where each time series sample is mapped into one label. The classification is performed by injecting prompt tokens into the task adapter, where the transformer processes the appended token sequence $ [p\_1, p\_2, …, p\_k, x\_1, x\_2, \ldots, x\_N] \in \mathbb{R}^{(N+k) \times D} $, where $k$ is the number of prompt tokens, $N$ is the number of input tokens, and $D$ is the transformer hidden dimension. We selected $k=5$ for all classification tasks, where the prompt tokens are averaged and passed into a linear projection head to obtain classification logits.
> > > > >
> > > > > **Anomaly Detection** We consider the unsupervised time series anomaly detection task. During the training stage, we minimize the reconstruction error of the normal data using MAE loss. In testing, anomalies are detected by comparing reconstruction errors to a threshold derived from the error distribution of the entire dataset. We employ event-based anomaly detection, and compute the detection accuracy for model evaluation. For all experiments, we sweep the anomaly rates across the data distribution {0.5\%, 0.75\%, 1.0\%, 1.25\%, 1.5\%} to compute the final rate for each dataset.
> > > > >
> > > > > **Imputation** We randomly generate and apply masks to the input to simulate and reconstruct the missing values. We evaluate imputation masking rates of 12.5\% and 25\%, with the mask ratio set to 1.5 times higher during training. The training objective minimizes the Mean Squared Error (MSE) between the ground truth and the model’s imputed outputs, focusing on both masked points and the entire series.
> > > > >
> > > > > **Regression** The regression task is performed exclusively on the synthetic dataset in a within-domain manner using a regression task adaptor during fine-tuning. The procedure is similar to the classification task, with the key difference being the loss function: classification uses cross-entropy loss, while regression employs MAE loss for fine-tuning.
> > > > >
> > > > > Note that all time series samples are normalized before encoding and denormalized after decoding following the existing pipeline.
> > > > >
> > > > > [3] Wu, H., Hu, T., Liu, Y., Zhou, H., Wang, J., & Long, M. (2022). Timesnet: Temporal 2d-variation modeling for general time series analysis. arXiv preprint arXiv:2210.02186.
> > > > >
> > > > > ## 4. Model specs.
> > > > >
> > > > > We would like to emphasize that model specs (detailed architectures) have been specified in Appendix section B.2.2. (line 1029-1036). All experiments are implemented in PyTorch and conducted using 2 NVIDIA Tesla V100 32GB GPUs.
> > > > >
> > > > >
> > > > > **Finally, we would like to thank all the reviewers for their comments and detailed feedback again. We believe that we have made a significant effort addressing your comments, which have improved the clarity and quality of our manuscript. We will answer individual questions on the individual responses, and we hope the response will address your concerns about our work.**

---

### Meta-Review · Area_Chair_DuRE · 2024-12-11

**Metareview:**

The paper proposes NoTS, a transformer-based method for pretraining time series data by reinterpreting them as sequences of functions at different levels of detail. This is achieved through "degradation operators" that progressively smooth the time series. NoTS uses an autoregressive transformer to predict the next, more detailed "function" in the sequence. The authors claim this approach leads to better generalization and performance on various time series tasks, supported by theoretical analysis and empirical results on classification, imputation, and anomaly detection.

Strengths of the Paper:
Novel Approach: NoTS offers a new perspective on time series representation using functional degradation and autoregressive prediction.
Theoretical Foundation: The paper provides a theoretical analysis to support its claims.
Clear Presentation: The paper generally presents complex concepts effectively.

Weaknesses of the Paper:
Limited Evaluation Scope: The lack of experiments on long-horizon forecasting is a major weakness, hindering a complete assessment of NoTS's ability to capture long-range dependencies, a key claim in the introduction.
Sensitivity to Degradation Operators: The paper lacks a thorough analysis of how the choice of degradation operators affects performance and provides limited guidance on their selection. This dependence on the choice of operators raises concerns about the method's robustness and generalizability.
Unclear Comparisons: The comparison to N-BEATS, a relevant prior work, is insufficient. A more detailed discussion of the similarities and differences is needed to highlight NoTS's contributions.
Missing Details: Reviewers pointed out a lack of clarity and detail in the model architecture, data flow, and specific implementation choices, making it difficult to fully understand and reproduce the work.

While NoTS presents a novel approach with some theoretical and empirical support, the weaknesses outweigh the strengths. The limited evaluation scope, particularly the absence of forecasting experiments, raises serious concerns about the validity of the authors' claims. The lack of analysis on the impact of degradation operators and unclear comparisons to existing work further weaken the paper.  Therefore, rejection is recommended.

**Additional Comments On Reviewer Discussion:**

The rebuttal period involved discussions focused on clarifying aspects of the paper and addressing concerns regarding the evaluation scope and the impact of degradation operators.

Key Points Raised by Reviewers:

tTyA: Requested analysis on the sensitivity to degradation operators and evaluation on long-horizon forecasting tasks.
DXpz: Pointed out missing related work, notational inconsistencies, and lack of clarity in model description.
WkQH: Highlighted the absence of forecasting experiments, limited ablation studies, and the need for clarification on technical details.
How the Authors Addressed These Points:

The authors performed additional ablation studies to analyze the impact of different degradation operators.
They conducted preliminary forecasting experiments on three datasets.
They added discussions of related work, clarified notation and model details, and provided more information in the appendix.
How the Reviewers Weighed in:

tTyA: Seemed satisfied with the authors' response.
DXpz: While appreciating the clarifications, they maintained a negative rating.
WkQH: Did not explicitly express their updated opinion.
My Assessment:

While the authors attempted to address the reviewers' concerns, the core weaknesses remain. The preliminary forecasting experiments are not convincing enough to compensate for the lack of thorough evaluation on this crucial task. The additional ablation studies, while helpful, do not fully address the concerns about the sensitivity to degradation operators.  The paper still lacks a comprehensive analysis and clear guidance on their selection.  Therefore, despite the authors' efforts, the paper does not meet the required standards for acceptance.

---

### Decision · Program_Chairs · 2025-01-22

Reject